

# The long-term variability of extreme sea levels in the German Bight

Andreas Lang[1,2] and Uwe Mikolajweicz[1]

[1]Max-Planck-Institute for Meteorology, Hamburg
[2]International Max-Planck Research School (IMPRS)

**Correspondence:** Andreas Lang (andreas.lang@mpimet.mpg.de)

**Abstract.**

We investigate the long-term variability of extreme high sea levels (ESL) in the southern German Bight and associated large-scale forcing mechanisms in the climate system using simulations covering the last 1000 years. To this end, global MPI-ESM simulations from the PMIP3 *past1000* project are dynamically scaled-down with a regionally coupled climate system model focusing on the North Sea.

We find that the statistics of simulated ESL compare well with observations from the tide gauge record at Cuxhaven but show large variations on interannual to centennial timescales. ESL arise independent of preferred systematic oscillations and are to a large extent decoupled from variations of the background sea level (BSL). Large scale circulation regimes associated with periods of high ESL are regionally consistent and similar to those associated with elevated BSL, but the location of the respective centers of action of the governing sea level pressure (SLP) dipole differs. While BSL variations correlate well with the wintertime North Atlantic Oscillation (NAO), ESL variations are rather associated with a dipole between northeastern Scandinavia and the Gulf of Biscay, leading to a stronger local north-westerly wind component in the North Sea. Potential links with solar or volcanic forcing are masked due to the high ESL variability.

The high internal variability stresses the irreducible uncertainties related to traditional extreme value estimates based on shorter subsets which fail to account for long-term variations. Existing estimates of future changes in ESL may be dominated by natural variability rather than climate change signals, thus requiring larger ensemble simulations to assess future flood risks.

*Copyright statement.* TEXT

# 1 Introduction

Inundation due to storm floods is one of the major geophysical risks in coastal regions and bears high damage potential for coastal environments, in both natural and socio-economic terms. This is especially important for low-lying regions such as coasts and estuaries of the Southern North Sea and, in particular, the German Bight. Situated between Denmark to the North and the Netherlands to the West, the German Bight is a shallow shelf sea under the influence of the major northern hemispheric storm-track paths. At the same time, the geometry of the resonator in combination with the shallow water depths of under approximately 40 meter (see Fig. 1) leads to relatively strong tides, with a maximum tidal range of around 4.5 m in the German





Bight. Thus, storms can generate particularly high floods in this region. The region has seen many devastating storm floods in the past; one of the most severe, the great storm flood 'Grosse Manndrenke' (the great man's drowning) in 1362 resulted in death tolls of tens of thousands, the destruction of numerous settlements including the disappearance of the legendary town 'Rungholdt' (Heimreich, 1819) and shifts in the Wadden Sea coastline (Hadler et al., 2018). More recently, the great flood in

1962, resulted in a high death toll and vast economic loss along the coastal regions of Germany and in particular the city of Hamburg. Disasters like these have driven extensive research in the field; yet, most studies focused either on individual events, on the observed trend during the last couple of decades, or a projection of future storm flood exposure. Variations on longer time scales and their underlying mechanisms have received less attention. However, a deeper understanding of the long-term variability of strength and occurrence of extreme storm floods can be of great importance for coastal planning and risk as-

sessment. Here we assess the longer-term variability of high sea level extremes in the German Bight using a novel regionally coupled model approach over a 1000-year long simulation period.

Following Pugh (1987), the sea level at a certain point and time can be decomposed into three factors: Meteorological surge, astronomical tide and background sea level (BSL). The *surge* is the "dynamic response of the sea surface to forcing by the

atmosphere" (Mawdsley and Haigh, 2016) and can consist of (i) the *local wind surge* that pushes water against the coast, (ii) an *external surge* generated in the North Atlantic by fast bypassing cyclones and air pressure variations that travel as a Kelvin wave counterclockwise in the North Sea (Gönnert et al., 2012), and (iii) the direct effect of air pressure acting on the water surface (the *inverse barometric effect*). The total sea level is thus a product of interactions between the surge component, the astronomical tide and the underlying longer term change in BSL, the latter of which depends on various oceanographic and at-

mospheric processes such as coastally trapped waves, local steric effects or longshore winds (Dangendorf et al., 2014a; Sturges and Douglas, 2011; Calafat et al., 2012). Extreme sea levels particularly arise when these components are in superimposition, e.g. if a strong wind surge occurs concomitantly with a tidal maximum, or – as a result of the tide-surge interaction – a few hours before on the rising tide (Horsburgh and Wilson, 2007). Topographic features, such as water depth, sand bars or reefs can further affect their height locally. These components interact non-linearly (e.g., Kauker and von Storch, 2000; Plüß, 2004) and

are non-stationary, with variations occurring on multiple timescales. Long term changes in any of those components, e.g. due to internal variability or external climate change, may substantially alter the risk associated with sea level extremes. Since we are solely interested in high water extremes, we refer with the term *extreme sea level* (ESL) only to the upper end of the distribution.

Many studies have investigated the recent dynamics of ESL both regionally and globally using data from tide gauges (e.g.,

Tsimplis and Woodworth, 1994; von Storch and Reichardt, 1997; Woodworth and Blackman, 2004; Marcos et al., 2009; Menéndez and Woodworth, 2010; Mudersbach et al., 2013; Weisse et al., 2014; Wahl and Chambers, 2015) or using barotropic surge models for hindcasts of storm-surges (e.g., Kauker and von Storch, 2000; Langenberg et al., 1999; Woth, 2005; Weisse and Plüß, 2006). For a review of past storm surge statistics and projected changes in the North Sea region see Weisse et al. (2012). While most studies agreed on an overall increase in storm surge activity along the German Bight since the 1960s

(e.g., WASA-group, 1998; Weisse and Plüß, 2006; von Storch and Woth, 2008; Mudersbach et al., 2013), both observations



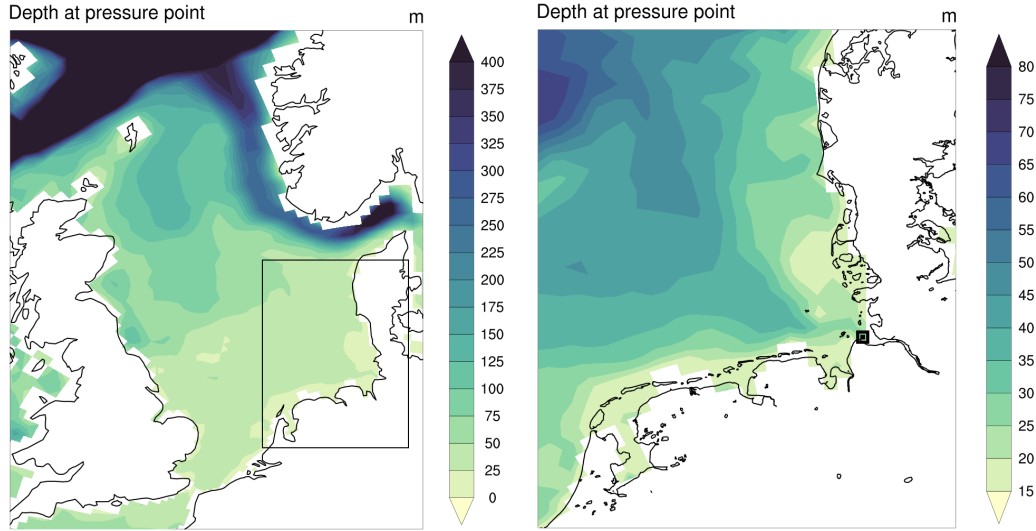

**Figure 1.** Bathymetry of the North Sea (left) and the German Bight (right; the main study location Cuxhaven is marked by the black square) as represented in MPIOM. The model's land mask is shown in white, the present-day coastline is shown in black.

and hindcast simulations over a longer time span have set this recent trend into the perspective of a marked multi-decadal variability during the last century (e.g., WASA-group, 1998; Dangendorf et al., 2013b; Mudersbach et al., 2013; Weisse and Plüß, 2006) with higher values at the beginning and end of the century.

The data record, though, is limited and only few high-frequency tide gauge records (e.g. Cuxhaven) date back more than
a couple of decades. Thus, conclusions on multidecadal to centennial variability as well as the separation of longer-term fluctuations from the transient sea level rise are difficult from a statistical point of view. Concerning the latter, different studies on German Bight sea level reported similar ESL and BSL trends (e.g. Kauker and Langenberg, 2000) or trends at rather different rates (e.g., Mudersbach et al., 2013; Dangendorf et al., 2014b) – a question of great importance for estimations of future ESL behavior on top of a gradual sea level rise. Further, as ESL are by definition rare, the attribution to modes of climate
variability, which often operate on similar or even longer timescales, is hampered by the short instrumental record. While many studies have related mean sea level variations to modes of climate variability, esp. the North Atlantic Oscillation (NAO) (e.g., Wakelin et al., 2003; Dangendorf et al., 2012; Ezer et al., 2015), the dominant pattern of atmospheric variability over the North Atlantic (Hurrell, 1995), which showed coherent trends during the last decades, only a few have set this in context with ESL variations (Woodworth and Blackman, 2004; Woodworth et al., 2007; Marcos et al., 2009; Marcos and Woodworth,
2017). Finally, such long-term ESL fluctuations can also have important implications for storm flood protection measures. The design of coastal defense structures in Germany is based on deterministic or statistical approaches (e.g., MfLR, 2012). For the latter, water levels with assigned return periods are needed, which are typically based on parametric extreme value analysis of observed sea level data. Yet, due to the relatively short tide gauge records, the quality of the estimation of return periods longer than the investigated sea level data series depend on the type of extreme value distribution and its goodness-of-fit to the



data. Additionally, any longer-term variability in ESL further complicates the estimation of high return levels, as they depend on the state of long-term variability during the underlying baseline period. Here we argue that significant centennial variations in high-impact return levels entail a large source of uncertainty for parametric ESL estimates. The standard approach using a typically short baseline period for such sea level estimates thus fails to reflect the possible range of most extreme events that

happen only once or twice during that period.

A longer, high-frequent data series as obtained from a climate simulation can offer a statistically more robust assessment of these problems, as the full time series can be treated as an ensemble of data series comparable in size to the observational record. However, currently available long-term climate simulations do not include tides and have an insufficient resolution to realistically represent storm surges in the North Sea. Dynamical surge models or regional climate models, driven by global

climate model simulations, can provide a better representation of small-scale processes, topographic influences and land-sea contrasts, and are thus better suited for the simulation of extreme events. Due to their open lateral boundaries, however, they cannot account for a consistent propagation of external signals into the study region, which has been shown to affect North Sea sea level variability (e.g., Chen et al., 2014). Here we employ a global ocean model with regionally high horizontal resolution, which allows a consistent simulation of signals propagating from the open Atlantic onto the North West European shelf,

coupled to an atmospheric regional model to dynamically downscale the climate variations from a *Last Millennium* simulation from MPI-ESM (Jungclaus et al., 2014; Moreno-Chamarro et al., 2017b). With a long-term simulation, this study allows a non-parametric approach in estimating such high return levels and can thus give insight into the uncertainties in extreme value analysis when based on short records. The goal is to describe and understand the (multi)decadal variability of ESL in the German Bight as well as their relationship with BSL and large-scale climate variability, which to our knowledge has not yet been

investigated in a model study over such long timescales.

This article is structured as follows: Section 2 introduces the model system and experimental setup. In section 3, we present results on ESL variability, including the validation of the model with respect to observations from tide gauges (3.1), the relation to BSL (section 3.2) and climate variability (section 3.3). The results on ESL variability as well as some implications

for observation-based extreme value estimates are discussed in section 4. Finally, in section 5 we close with a summary and conclusions for coastal defense measures.

## 2   Methods

### 2.1   Model system and experimental design

This study employs a regionally coupled climate system model, consisting of the global ocean model MPIOM (Marsland

et al., 2004; Jungclaus et al., 2013) and the regional atmospheric model REMO (Jacob and Podzun, 1997). Both models are interactively coupled over the wider EURO-CORDEX domain (e.g., Jacob et al., 2014) with the coupler OASIS-3 (Valcke, 2013). The coupled model system has been described in Mikolajewicz et al. (2005); Elizalde et al. (2014) and Sein et al. (2015); a sketch is shown in Fig. 2.




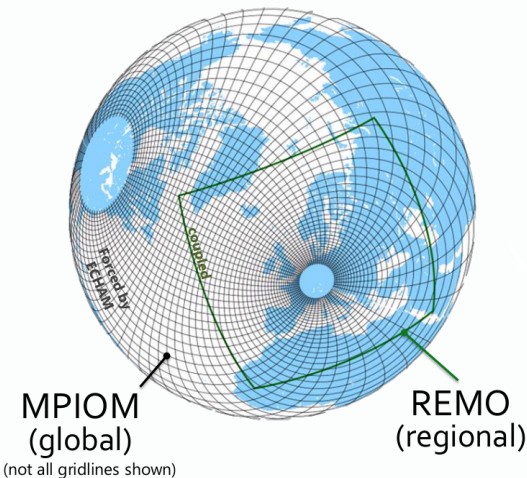

**Figure 2.** Coupled model setup, consisting of MPIOM (black) and REMO (green).

REMO is run with a 0.44 degree setup, corresponding to approx. 50 km grid spacing, and with 27 vertical levels covering Europe, northern Africa and the northeast Atlantic. MPIOM is run on a stretched grid configuration with a nominal horizontal grid resolution of 1.5° and 30 vertical layers. In order to maximize the grid resolution in the study focus area, the model's poles are shifted to Central Europe and North America, respectively. This results in a maximum grid resolution of under 10 km in the

German Bight and thus enables a more realistic simulation of small-scale shelf processes. Further, it includes the full luni-solar ephimeridic tidal potential according to Thomas et al. (2001). MPIOM's uppermost layer thickness is 16 meter. At the same time, the ocean model's global domain without lateral boundaries allows the full simulation of signals propagating from the open Atlantic into the North Sea. This model setup is identical to the one used in Mathis et al. (2018).

We employ this model setup to downscale transient coupled climate simulations performed with the paleo-version of the

global Earth System Model climate model MPI-ESM (Max-Planck-Institute Earth System Model) in its low resolution (LR) version (Giorgetta et al., 2012), corresponding to 1.875° or approx. 200 km grid spacing in the atmosphere. The parent global simulations cover the period 900–2000AD and comprise parts of the PMIP3 simulation *Last-Millennium* (850–1850AD, (Jung-claus et al., 2014; Moreno-Chamarro et al., 2017b)), extended with the corresponding CMIP5 "historical" simulation (1850-2005AD (Taylor et al., 2012)), including all relevant transient forcings. Greenhouse gases (GHG) follow PMIP3 protocol

(Schmidt et al., 2012), solar irradiance is prescribed after Wang et al. (2005) and volcanic eruptions in terms of radiation im-balance after Crowley et al. (2008) (see Supplementary Fig. A4). 6-hourly atmospheric forcing derived from the atmosphere component of the driving GCM is used as lateral boundary conditions for REMO, or as surface forcing for MPIOM outside the coupling domain, respectively. Topography and coastlines as well as ice sheets are constant, and thus transient sea level modulations due to ice sheet melt or changes in coastal morphology including land sinking or lifting are not accounted for.

The downscaling was performed as one continuous run with hourly coupling started in year 900AD, the first 100 years are used as a spin-up. Sea level and selected atmospheric fields are stored at hourly resolution. Additionally, in order to





quantify the variability from the downscaling process, as well as the contributions of natural variability and external forcing, we performed two further downscalings, one of the same global realization and one of another ensemble member of the global *Last-Millennium* simulations, each starting in 1400AD, with the first 100 years again used as sin-up (see Supplementary Material A3). The implications of these additional downscalings are discussed in the respective sections; yet, for simplicity,

we only show results from the 1000-year continuous downscaling.

## 2.2  Extreme value sampling

Several techniques have been used to characterize extreme sea level, e.g. the use of high percentiles (e.g., Woodworth and Blackman, 2004; Dangendorf et al., 2013a), the selection of *r-largest* maxima over a block of time (e.g., Araújo and Pugh, 2008; Méndez et al., 2007; Marcos et al., 2009), and the selection of peaks over a certain threshold (*POT*) (e.g., Méndez et al.,

2006). While the choice of the respective percentile, threshold, block length or number of block maxima is essentially arbitrary, the resulting events are sensitive to the choice of extreme value sampling index which represents a trade-off between bias (too high $r$ or too low threshold) and variance (too low $r$ or too high threshold) of the estimates.

Here, we have chosen the annual maximum sea level as an index representing ESL. Due to its relative definition of what constitutes an extreme it is robust to temporal variations. The use of a 'direct' method for ESL (instead of e.g. *surge residual*

or *skew surge*) is chosen in order to avoid a decomposition between tidal and surge parts and their nonlinear interaction. Since storm floods primarily occur in winter, annual statistics are computed for years defined as starting in July and ending in June in order to not split up one storm flood season. If not specified otherwise, these definitions are used when referring to ESL in the text.

For the design of coastal defense structures, policy makers and adaptation planners often require statistics of water levels of

a certain assigned return period (e.g. (MfLR, 2012)), especially those of high impact but low probability, i.e. the upper tail. The return periods and associated exceedance probabilities are typically estimated based on parametric extreme value analysis of the available instrumental data record. As data records rarely date back more than a couple of decades, this implies a substantial extrapolation of the data. That is, in order to obtain estimates for large return periods, different extreme value distributions are fitted to the comparably short data. The choice of extreme value distribution depends on the considered extreme value sampling

method. While the *POT* method is linked to the Generalized Pareto distribution, the $r$-largest samples follow approximately a three-parameter generalized extreme value (GEV) distribution (e.g., Coles et al., 2001). Its cumulative distribution function is

$$F(z; \mu, \sigma, k) = \begin{cases} e^{-(1+k(\frac{z-\mu}{\sigma}))^{-1/k}} & k \neq 0 \\ \\ e^{-e^{-\frac{z-\mu}{\sigma}}} & k = 0 \end{cases} \tag{1}$$

where $F$ is the probability that a water level $z$ will not be exceeded, while $\mu$, $\sigma$, and $k$ are the location, scale and shape parameters, respectively. The special case for $k = 0$, $k < 0$ and $k > 0$ represent three extreme value families, namely the

Gumbel (type I), Weibull (type II) and Frechet (type III) distribution (Coles et al., 2001). From Eq. 1, the probability of exceedance is $E = 1 - F$, where $E(z)$ represents the expected frequency of events exceeding $z$. The expected time-interval





between events of level $z$ or greater, the return period $RP$, is

$$RP(z) = 1/E(z) \tag{2}$$

However, the choice of extreme value distribution, and thus the extreme value sampling method, is not trivial and ultimately represents a tradeoff between bias and variance of selected extremes.

## 3 Results: Extreme sea level variability

The simulated timeseries of ESL at Cuxhaven (see Fig. 1) over the last 1000 years is shown in Fig. 3 (black curve). For comparison, we show the frequency of storm floods as events binned per decade (blue), following the storm surge definition from the Federal Maritime and Hydrographic Agency (BSH) (Müller-Navarra et al., 2003). Heavy (extreme) storm floods correspond to elevations above 2.5 (3.5) meter, relative to the long-term mean high water level (MHW). The BSL as a reference in terms of winter median is shown in green. Sea level is given in meter above the long-term mean, the model location of Cuxhaven refers to its nearest gridpoint.

Simulated ESL range between 2 and 5.5 m above the long-term mean. With a standard deviation of 50 cm and a maximum year-to-year amplitude of roughly 3 m, ESL exhibit large interannual variations and pronounced variability on various timescales. Yet, the highest events occur without pronounced clustering throughout the full 1000 years. This variability is analyzed in more detail below.

### 3.1 Validation of simulated storm surge statistics

As the reality can be viewed as only one realization of the climate system one cannot compare individual historic events with the simulation. Yet, a comparison in terms of extreme value statistics is possible. To validate the model's performance considering storm flood statistics, we compare the simulated ESL with observations from the tide gauge record in the German Bight (data from AMSeL project, see Jensen et al. (2011); Wahl et al. (2011)), and specifically show results at Cuxhaven which, starting in 1900 marks one of the longest reaching records of the German Bight tide gauge stations. The long-term trend in the tide gauge data has been removed.

While the general North Sea tidal oscillations are well reflected in the model, the tidal range is underrepresented at Cuxhaven (see Supplementary Fig. A2 for a comparison of the broader highest water level from observations and model simulation, respectively). Accordingly, this results in a lower long-term MHW. Relative to the respective long-term MHW, however, simulated and observed ESL compare well, and we therefore express ESL in such relative terms in the remainder of the study.

Figure 4 shows return values of ESL compared to the Cuxhaven tide gauge record. In order to better compare the different data record lengths, the simulated 1000 years of data have been split into ten 100 year-long segments, roughly the length of the Cuxhaven tide gauge record. Both slope and magnitude above MHW are well-captured; the return values inferred from observations data lie at all periods within the spread of the model simulation. Yet, other than in the simulation, observations tend to level off for return periods above 20 years, before they rise again for return periods higher than 50 years, while simulated



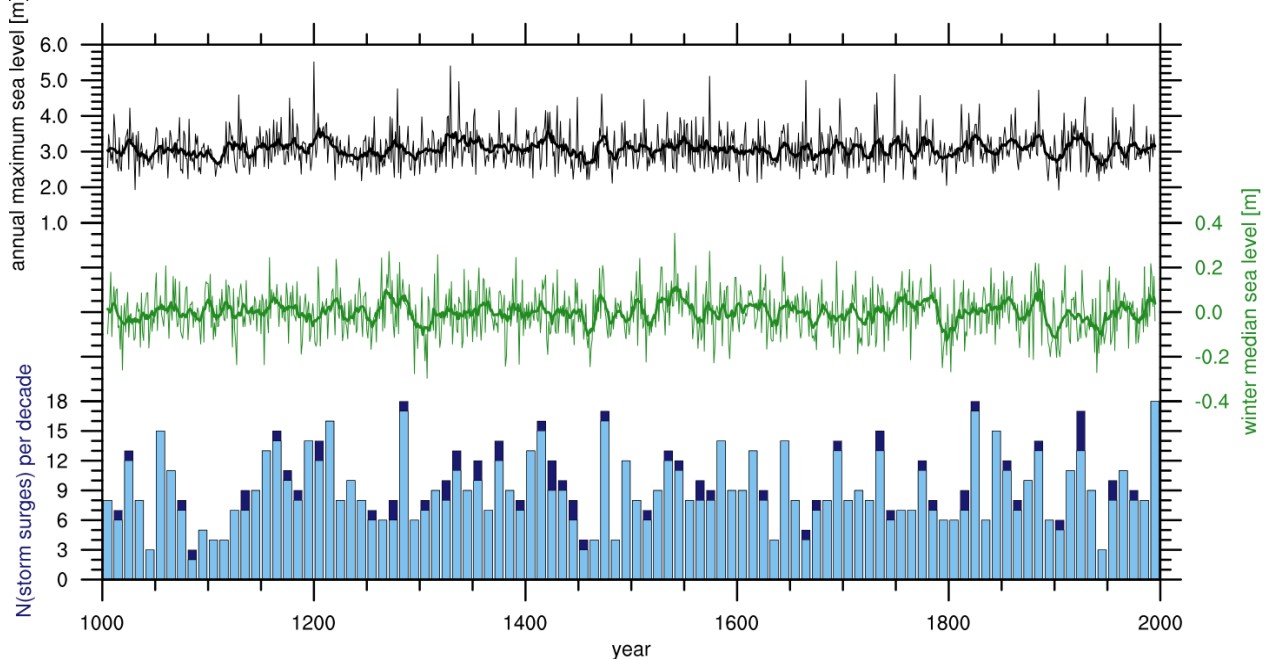

**Figure 3.** Simulated ESL (black), winter median sea level (green) as well as number of heavy (blue bins) and extreme (dark blue bins) storm surges per decade at Cuxhaven. Thick lines denote the 11-year running mean.

return levels rather increase steadily. As a consequence, the 100 year return levels ($RL_{100}$) in all but the 11th century exceed the corresponding return water level inferred from the instrumental record. The large scatter of about 1.2 m between the highest simulated sea levels of each century has important implications for extreme value analysis (see section 4).

5  The spatial structure of simulated return values (Fig. 5) shows lowest values for open waters which increase towards the coast, especially the inner German Bight. Most points along the German Bight coast (circles represent Cuxhaven, Husum, Norderney and Delfzijl (Netherlands)) compare well with the respective tide gauge records. Yet, while the return values at Cuxhaven lie slightly higher than the observed ones, ESL along the coastline of Lower Saxony and the Netherlands appear rather underrepresented, pointing to a bias towards too zonal (westerly) winds. Furthermore, the model's minimum water depth is with 16 meter well above the shallow shorelines of the Wadden Sea (see Fig. 1) and thus likely leads to lower surge heights.

10  Note that for Husum and Norderney the observational record does not date back the full 100 years, so these points are only shown for the 50 year return period.

The seasonality is well-captured, with strongest and most frequent storm floods in winter (Supplementary Fig. A3), especially in the months of October-January. However, the distribution is slightly shifted towards autumn and early winter.



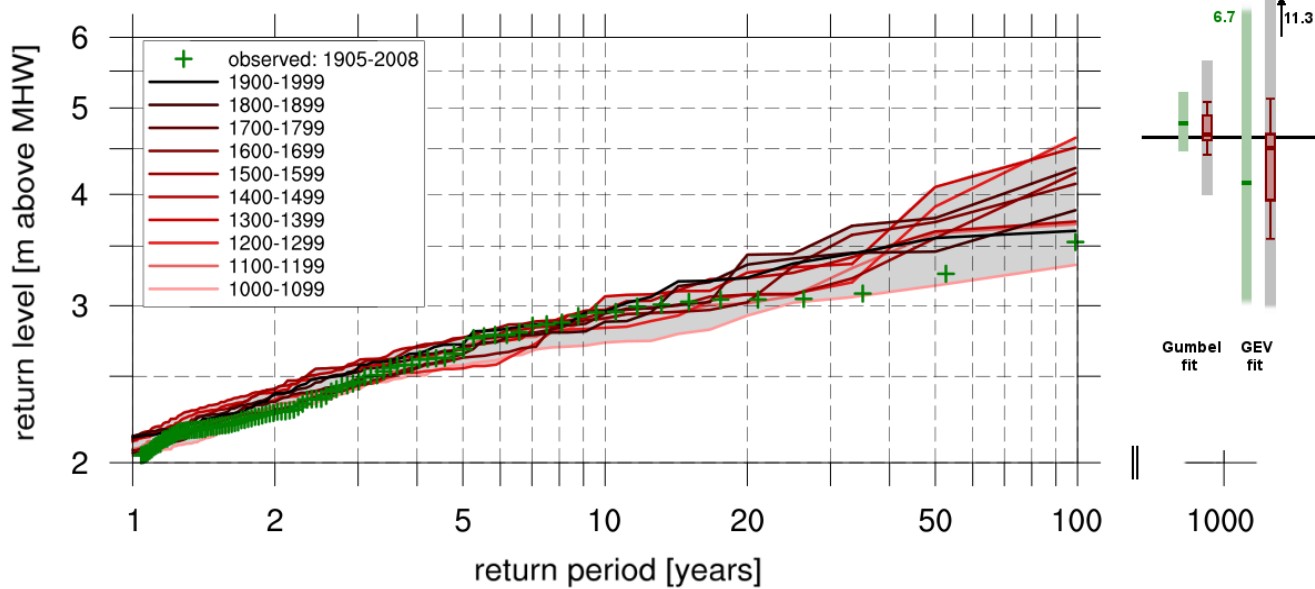

**Figure 4.** Return value plot using plotting positions of simulated sea level at Cuxhaven [m over MHW] (colored lines representing 100-year long subsets of the full 1000 years) against observations from tide gauges (green crosses). The bars on the right mark the corresponding $RL_{1000}$ estimates using a Gumbel (left) and GEV (right) fit to the observations (dark green, 95% confidence interval in light green) and to each 100-year subset of the simulation (red Box-Whiskers, range of the 95% confidence intervals in grey). The horizontal black line shows the $RL_{1000}$ directly inferred from the full simulation.

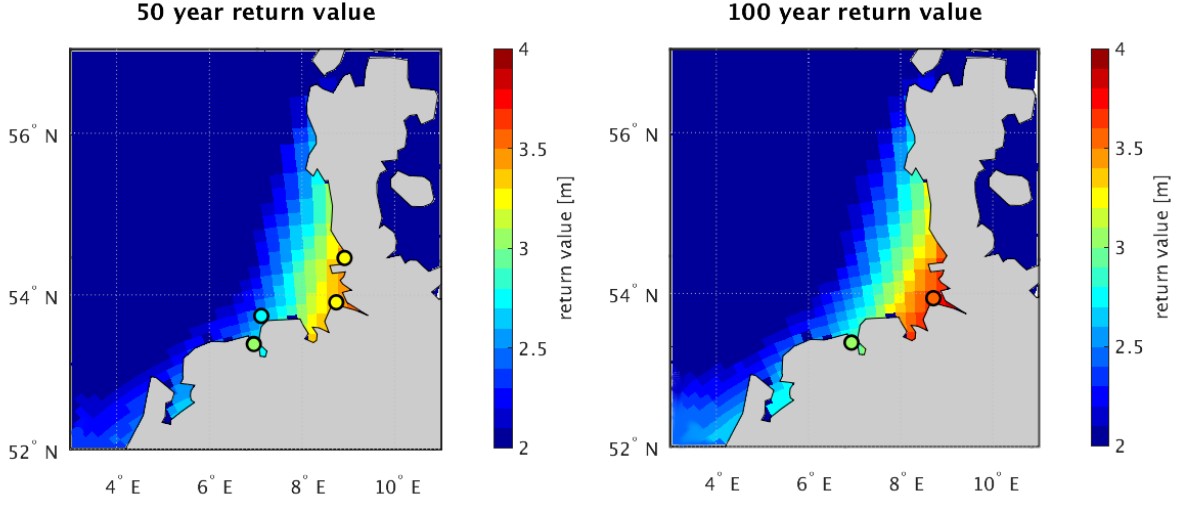

**Figure 5.** *Gridded 50 and 100 year return levels and the corresponding water values from tide gauge observations at selected locations (circles). All values given in m above mean high water.*



In agreement with observational studies (Gerber et al., 2016), simulated storm floods at Cuxhaven stem from predominantly west-north-westerly directions, while their associated daily pressure anomaly patterns are similar to observations of storm flood weather situations (Donat et al., 2010; Dangendorf et al., 2014c).

We thus conclude that the model reasonably reproduces storm flood physics and statistics. Due to the good skill in repro-
ducing storm surge statistics as well as the comparability owing to its relatively long instumental record, the remainder of this study will focus on Cuxhaven only. However, the temporal variability of other gridpoints along the German Bight is very similar (see Supplementary Fig. A7) and the here discussed results qualitatively agree irrespective of the exact location.

## 3.2   Relation to the background state

An important question concerning ESL variability as well as future ESL projections is the relation to time-averaged sea levels,
i.e. the background state. Separation into the different components of extreme water levels, such as the subtraction of mean and tide are useful methods to investigate this question (e.g. Woodworth and Blackman, 2004). Reviewing literature on recent ESL trends globally, Woodworth et al. (2011) concluded that the majority of studies suggest an increase over the last century, but at most locations at rates comparable to those observed in BSL. However, analyzing tide gauge records in the German Bight, Mudersbach et al. (2013) found differences in linear trends in high sea level percentiles from those in mean sea level. Are these
differences representative for this period only or can the finding be extended to a general statement?

Fig. 3 shows both ESL in terms of annual maxima (black) and BSL in terms of winter median sea level (green), and their respective 11-year running mean. We choose the median instead of the more simple mean in order to not obtain a skewed value due to an exaggerated influence of the very maxima. As the predominant storm surge season we only average over an extended winter period (October - March). Neither ESL nor BSL exhibit long-term trends, but show high interannual to multidecadal
variability. Yet, their modulations are not always coherent: As the histogram at the bottom of Fig. 3 shows, years with one *extreme storm surge* do not necessarily coincide with a greater occurrence of more moderate *heavy storm floods* or elevated BSL. The correlation between BSL and ESL is comparably low ($r = 0.35$) and highly variable over time (see black curve in Fig. 6 for a 100 year running correlation), while the different magnitudes of variances lead to a low explained variance ($R^2 = 0.12$). While there are periods of significant positive correlation between BSL and ESL after 1400, lower insignificant
correlations are prevailing during the 1st half of the last millennium. That is, the coherent behavior between mean and upper-end extreme sea level states varies on decadal to centennial timescales. After subtraction of the annual median from the ESL, the correlation between the resulting atmospheric surge residual and BSL further reduces and is insignificant over a large fraction of the last 1000 years (blue curve in Fig. 6, $r = 0.10$, $R^2 = 0.01$). In fact, significant coherence only applies in the 15th century and towards the end of the past millennium on timescales of several decades. This further indicates that the similar
trends between ESL and BSL during the last century that have often been described (Kauker and Langenberg, 2000; Menéndez and Woodworth, 2010) might merely be an unusual state if compared to a longer time horizon as obtained from our long-term simulation.

Are the temporal modulations of coherent behavior between ESL and BSL due to different modes of variability and are there any systematic variations in ESL? Spectral analysis (Fig. 7a) shows that ESL exhibit white power spectra across all resolved




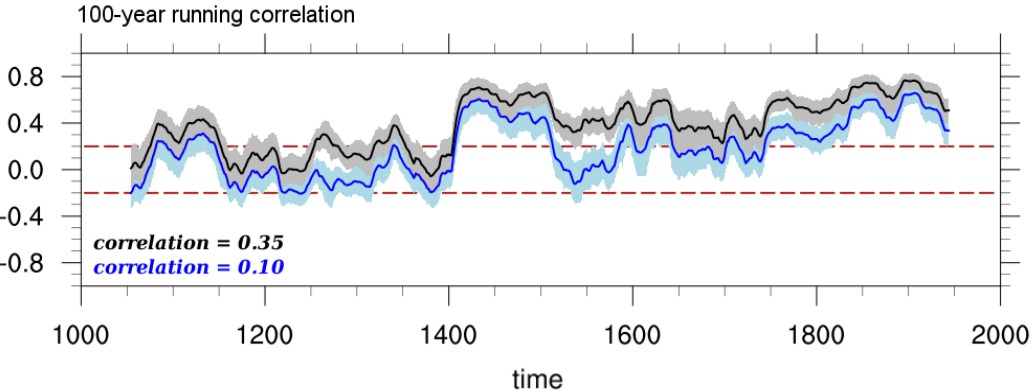

**Figure 6.** 100-year running correlation between BSL and ESL (black) and between BSL and median-reduced ESL (blue) with shading marking the uncertainty of the correlation using bootstrapping. Time series have been smoothed with a 11y moving window. Red dashed lines mark significant correlation on the 95th percent confidence level; The long-term correlation is given in the bottom left corner.

periods ($p = 0.57$ in Ljung-Box Q test) and do – except for a minor spectral peak around 8 years – not show preferred modes of variability (Fig. 7a). There is no significant difference between sites located along the coast of Lower Saxony and on sites at the coast of Schleswig-Holstein (not shown). On the other hand, BSL in terms of annual median sea level (Fig. 7b, separately shown for both winter and summer seasons) exhibits a red spectrum with more power on multidecadal and centennial timescales. The
predominance of lower frequencies in the power spectrum stresses the influence of the ocean which carries the "memory" of the system. In summer and further off the coast (not shown), the high-frequency variability is reduced and thus lower frequencies are dominating the power spectrum. In winter, the higher frequencies are not as damped and the spectrum appears flatter, as the wind stress effect on the water surface is more pronounced during winter when winds are strongest. Coherent variability of BSL and ESL is only visible on multidecadal timescales, at higher frequencies the random ESL variations outweigh the BSL
ones and do thus not result in coherent variations for periods shorter than several decades (Supplementary Fig. A8).

### 3.3    Relation to climate variability

Several mechanisms have been related to sea level variations, which mainly focused on those of the mean state. These range from large-scale atmospheric circulation patterns (e.g., Wakelin et al., 2003; Woodworth and Blackman, 2004; Chafik et al., 2017) over longshore winds and resulting Kelvin waves (Sturges and Douglas, 2011; Calafat et al., 2012) to steric variations
due to temperature oscillations (Frankcombe and Dijkstra, 2009). Mechanisms leading to longer-term ESL variations, however, remain more uncertain as data is limited, which challenges the robustness of statistical relationships between sea level extremes and other variables in the climate system. Yet, the patterns of large scale climate variability over the North Atlantic that potentially influence ESL may be different to those responsible for BSL variations. In order to investigate large-scale climate patterns associated with enhanced storm surge activity, we relate such periods with multi-decadal climate variability, both
internally generated as well as externally forced.



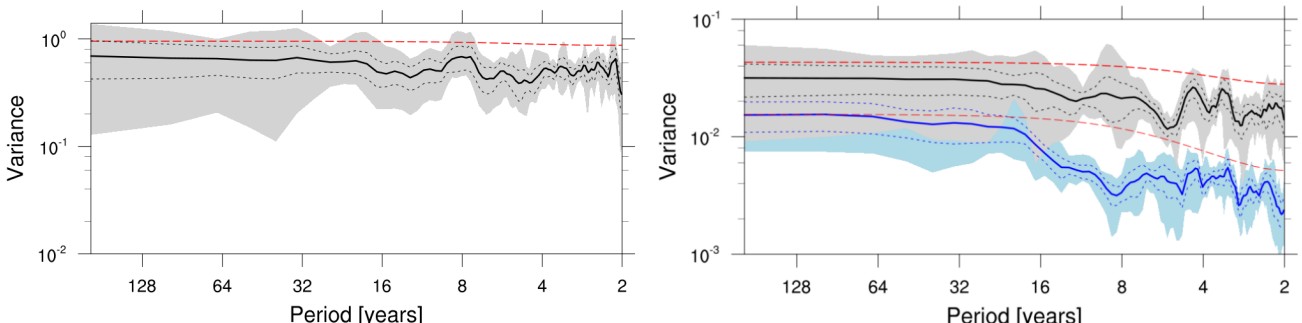

**Figure 7.** Power spectrum of sea level at Cuxhaven: Annual maximum sea level (left) and annual median sea level (right), split into winter (black) and summer season (blue). Spectra have been smoothed over 7 spectral estimates using a Daniell window (Daniell, 1946). The thick lines correspond to an average over 9 overlapping 200-year subsets of the full time series, the shadings mark their range. Red lines indicate the 95 % confidence bounds using a theoretical Markov spectrum (red noise), black dashed lines the 95% confidence bounds derived from the 9 realizations.

### 3.3.1 Internal variability

The NAO has often been linked to BSL variations in the North Sea (e.g., Wakelin et al., 2003; Woodworth and Blackman, 2004; Dangendorf et al., 2012), both through baroclinic as well as barotropic processes (Chen et al., 2014). Correlating observed ESL from tide gauges with climate reanalysis data, some authors found the same large-scale patterns responsible for high ESL as

the pattern persists after removing the annual median (e.g., Woodworth et al., 2007; Marcos and Woodworth, 2017)). Yet, the standard NAO is not necessarily the most indicative index: Kolker and Hameed (2007) have shown that the location of the centers of action comprising the NAO is affecting observed mean sea level trends and variability. Introducing a "tailored NAO index", Dangendorf et al. (2014c) showed that slightly different pressure constellations than the standard NAO can better describe the observed ESL variability in the German Bight. Additionally, other teleconnection patterns such as the East Atlantic

Pattern (EAP) or the Scandinavian pattern (SCA) have been shown to exert some influence on North Sea storminess (Seierstad et al., 2007) and mean sea level (Chafik et al., 2017). However, as some of the modes of climate variability are operating on similar timescales as the high-resolution instrumental sea level record, it is difficult to obtain robust conclusions.

As an indicator of the large-scale circulation, we compute positive composite maps of winter (Oct-Mar) mean sea level pressure (SLP) during times of high ESL at Cuxhaven. Composites have been calculated as the average over periods where

the simulated ESL time series at Cuxhaven exceeds its mean plus 1.5 times its standard deviation. The choice of the threshold is arbitrary, but the pattern remains robust to minor changes in its value. Again, we restrict the analysis to the extended winter season as storm surges primarily occur during these months. Further, SLP has been low-pass-filtered with a 3 year moving average to better investigate variability on longer timescales, but to still be able to account for the slightly pronounced variations with periods of around 8 years.





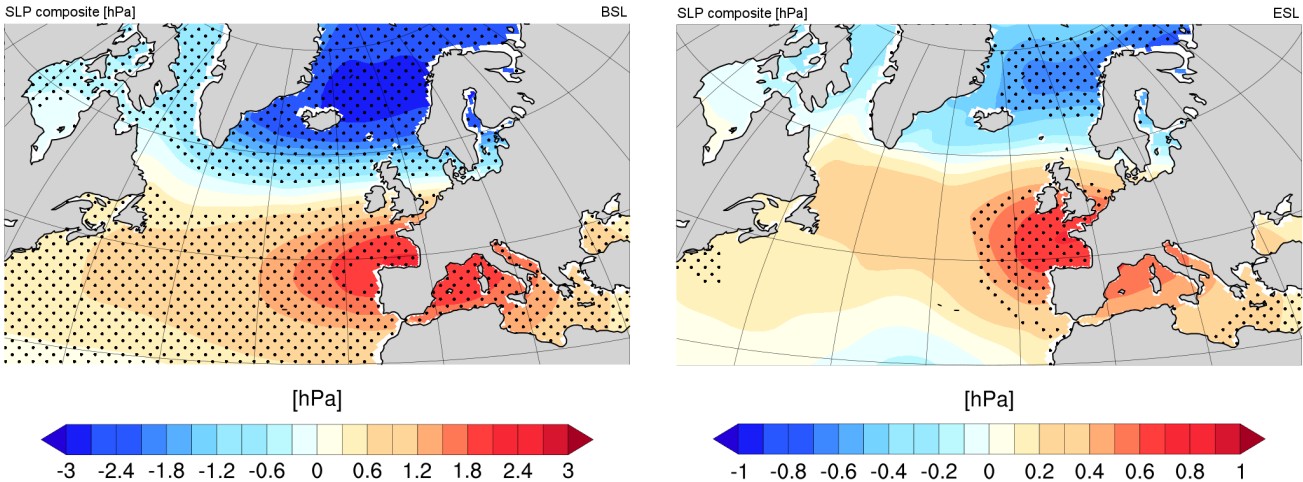

**Figure 8.** Composite gridded winter SLP anomaly for periods of high BSL (left) and ESL (right) at Cuxhaven. Stippling marks areas significant at the 5 % confidence level.

The SLP pattern associated with enhanced BSL (Fig. 8a) deviates from the long-term winter mean pattern as the meridional pressure gradient over the North Atlantic is strengthened, comprising a negative SLP anomaly East of Iceland and a positive one over the Iberian Peninsula. This dipole with a meridional axis is similar to a positive NAO and the correlation coefficient between BSL and NAO, computed as the leading principal component of the North Atlantic SLP, is significant ($r = 0.5$). This marks a qualitative agreement with the literature outlined above.

The SLP constellation favoring high ESL (Fig. 8b), however, differs slightly. As for BSL, it comprises a dipole over the northeast Atlantic, yet its centers of action are shifted to northeastern Scandinavia and the Gulf of Biscay, leading to a further eastward stretching Icelandic Low and a clockwise turned dipole favoring a more northwesterly wind component. This pattern is different to the meridional NAO-like dipole as in the case of elevated BSL. Due to the large ESL variability and the considerable noise in the extreme value time-series, the composite pattern is weaker pronounced than the one associated with high BSL. Note that the SLP composites are averaged over the winter season while the ESL time series is based on winter maximum values; the SLP pattern do therefore not represent the situation during individual extreme storm floods but rather reflect a general circulation pattern during times of enhanced storm surge activity. The anomaly structure resembles the dipole described by Dangendorf et al. (2014b) for cross correlations of SLP with observations of daily wind surges at Cuxhaven and agrees well with the mean weather situation triggering strong storm surges found by Heyen et al. (1996) in the wider region using statistical downscaling. Its spatial structure points to an influence of the Scandinavian Pattern in its negative phase (SCA−) onto the NAO centers of action, such as described by Chafik et al. (2017) for North Sea sea level variability. The pattern is also in agreement with a complementary analysis performed on the 2-6 day band-pass-filtered pressure variance (not shown), which indicates enhanced storm track activity over the northeast Atlantic and North Sea.

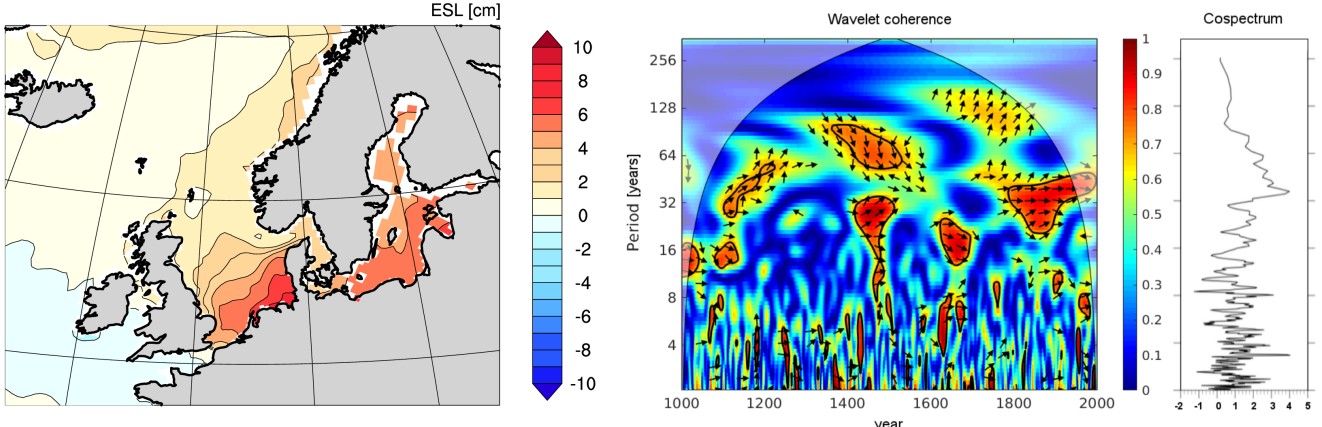

**Figure 9.** *Left:* Pointwise regression of the from Fig. 8 derived SLP index onto annual maximum sea level (color shading, [m per unit of SLP index]). *Right:* Wavelet coherence and cospectrum between annual maximum sea level at Cuxhaven and the tailored SLP index. Arrows to the right (left) indicate a positive (negative) correlation and upward (downward) arrows indicate a lag (lead) of the first time series. Thick contours designate the 5% significance level against red noise, the cone of influence is shown in a lighter shade.

As the SLP pattern associated with enhanced storm surge activity at Cuxhaven differs from the NAO (correlation of NAO and ESL: $r = 0.19$), we use the shifted centers of action (Biscay – Northeast Scandinavia) to define a SLP index based on the normalized SLP difference between those points of the dipole, to directly investigate the influence of this circulation pattern on storm surge activity in the wider region. Regressing the time series of this index onto the ESL field shows highest values not

only in the German Bight (Fig. 9a; correlation $r = 0.31$ at Cuxhaven), but also in the southern Baltic Sea where the regressed annual maximum sea levels are of similar magnitude. This suggests that periods of enhanced ESL activity in both German Bight as well as the southern Baltic Sea are linked via the same large scale circulation pattern. The spatial coherence of long-term ESL variations (see also Supplementary Fig. A7) is in agreement with the study by Marcos et al. (2015) using data from tide gauges globally. The wavelet coherence – which can illustrate both timescales and timing of coherent behavior of both

data series – between this index and ESL at Cuxhaven shows that this relationship acts mainly on multidecadal to centennial timescales (Fig. 9b).

The second downscaling (1500-2000) of the same GCM simulation yields a similar spatio-temporal variability. This indicates that the large-scale pattern responsible for high ESL activity is also a feature of the parent global simulation, which determines the temporal ESL variability on larger scales – the regionalization however can give added value in the development of dynamic

systems such as blockings and is more important for the specific surge heights and finer regional differences that are related to the exact wind direction and strength.



### 3.3.2 External forcing

It has been suggested that external influence, such as solar variations or large volcanic eruptions can have an impact on magnitude and phasing of various climate phenomena such as the Atlantic Multidecadal Oscillation (AMO) (Otterå et al., 2010; Knudsen et al., 2014), longer-term anomalous temperature regimes (e.g., Miller et al., 2012) or atmospheric variability patterns

such as the NAO (Swingedouw et al., 2011; Zanchettin et al., 2013), and can even trigger periods of enhanced storminess and coastal flooding (Barriopedro et al., 2010; Kaniewski et al., 2016; Martínez-Asensio et al., 2016). For instance, using geological proxy data of the central Mediterranean Sea, Kaniewski et al. (2016) argue for long-term correlations on cycles of around 2200-yr and 230-yr between storminess and solar activity; periods of lower solar activity will intensify the risk of frequent flooding in coastal areas. For the same region, Barriopedro et al. (2010) and Martínez-Asensio et al. (2016) also found coherent

decadal changes in solar activity and autumn sea level extremes from tide gauges linked to the 11 year solar cycle through modulation of the atmospheric variability, namely a large-scale wave train pattern, implying an indirect role of solar activity in the decadal modulation of storm flood frequency.

The last millennium has seen substantial variations in solar irradiation, that have affected surface temperatures and lead to various longer-term temperature regimes such as the Late Maunder Minimum (1675-1710) or Dalton Minimum (1790-

1840). Additionally, volcanic eruptions can alter the radiation balance substantially for a shorter time and clearly outweigh the variations of solar irradiance alone (see Supplementary Fig. A4). Due to the different timescales, magnitude and expected lag of a potential response to these external variations, it can be useful to separately investigate both forcings and their potential relationship with ESL variations through atmospheric variability.

However, a relation to extreme storm surge activity in the German Bight is not evident in our simulations. Wavelet coherence

between total solar irradiance and ESL at Cuxhaven (Supplementary Fig. A10a) and a 'superposed epoch' analysis between volcanic eruptions and ESL (Supplementary Fig. A10b) do not show a consistent significant relationship. Furthermore, the different temporal variations of ESL found in the downscalings of the two different global *Last Millennium* simulations (Supplementary Fig.A5) stresses the dominance of natural variability in the timing of ESL variations. Due to the high internal variability of ESL, any signal from a potential external influence is masked and there is no evidence of coherent variability

between German Bight storm surge activity and insolation variations – with or without the inclusion of volcanic forcing – during the last millennium. That is, extreme storm floods have occurred independent of major changes in forcing mechanisms or resulting long-term anomalous temperature regimes. This is in agreement with the findings by Fischer-Bruns et al. (2005) for multi-century simulations of mid-latitude storms. The above described link to atmospheric modes rather stresses the internal component of storm flood variability; this is in accordance with Gómez-Navarro and Zorita (2013) who have shown

that the decadal variability of atmospheric modes such as the wintertime NAO is mainly unforced in CMIP5 *Last Millennium* simulations.



### 3.4 Relation to ESL components

Which components are governing ESL variability? As described above, the high-end extreme sea levels arise as a combination of three components: the tide, longer-term base level variations and the surge residual, comprising all faster meteorological and oceanographic influences, from both local as well as remote forcing.

Following Woodworth and Blackman (2004), we investigate the surge residual via removal of the two other components, namely tide and background state in terms of the winter median. Removal of tides (using matlab program "t-tide" (Pawlowicz et al., 2002)) alters the extreme storm surge time series by up to 1 m, depending on the tidal phase during the storm surge. Yet, after removal of tides and median, the general features of variability and spectra remains similar (Supplementary Fig. A9a), stressing that the variability of extreme storm surges mainly stems from the atmosphere. This may not be surprising, as wind stress is expected to be the most important factor in shallow seas. The unchanged variability also implies that the absolute ESL index using annual maxima is a reasonable indicator of storm flood variability. Furthermore, the large-scale circulation pattern associated with high ESL qualitatively persists if annual median and tide are subtracted (not shown), although weaker. This is in accordance with findings by Woodworth and Blackman (2004) and Marcos and Woodworth (2017) who concluded that relationships with larger scale climate variability, and specifically the NAO, remain even after removal of the other storm surge components.

### 4 Discussion

The pronounced ESL variability on various time scales found in our simulation has important implications for the interpretation of the instrumental record, including trends as well as estimates of present and future storm floods. With a single realization of limited length such as the instrumental ESL record, statements about potential correlation with BSL, climate variability or alleged trends are statistically problematic and should therefore be treated with caution. For instance, setting recent ESL trends from the observational record ($5.7 \pm 4.3$ cm/decade for the 99.9th percentile of hourly sea level at Cuxhaven from 1953 to 2008, see Mudersbach et al. (2013)) in context with the simulated ESL variability shows that the trends lie within the internal variability obtained from the long-term simulation: Using a running trend with a window length similar to the aforementioned observational data (55 years) over the 1000 years of simulated data, a trend of the same or higher magnitude occurs in roughly 10% of the segments. A trend higher than the given upper uncertainty limit (10cm/decade) still occurs 10 times during the 1000 years, or in around 1% of the segments.

Further, the large variations in high return values (around 1 meter for $\mathrm{RL}_{100}$) illustrate the peril of using the standard approach of estimating extreme sea levels for flood protection standards. Typically, such estimates are based on parametric extreme value theory which is based on only a couple of decades of data (typically 30-50 years), be it the observational record or simulated data at end of the century. The effect of fitting different extreme value distributions onto the same baseline period alone can be significant: A recent study by Wahl et al. (2017) quantified the uncertainties related to different extreme value



estimates and showed that the "high-impact-low-probability" sea level states can vary substantially depending on both extreme value distribution and sampling technique. If for example a GEV distribution is fitted to a sample with more weight on more moderate extremes it might lead to a mismatch in higher return levels. For instance, Arns et al. (2013) have shown that for Cuxhaven, the use of $r$-largest order statistics with $r > 1$ value per year leads to an overestimation of return water levels.

Together with the considerable spread of high-impact return level statistics however, this results in even larger uncertainties: The error bars in Fig. 4 illustrate the uncertainties related to extrapolation from shorter subsets. The 1000-year return level ($\mathrm{RL}_{1000}$) estimated from the 100-year long instrumental record yields error bars of 0.8 (3.2) meter for a Gumbel (GEV) fit using maximum-likelihood estimation. The uncertainty is further stressed by the ensemble of ten 100-year long segments, which in itself scatter around 1.2 meter for 100-year return levels. This means in turn that a flood protection standard based on

the highest observed sea level from a 100 year data set could as well correspond to a 30 year return level, only if another 100 year period were considered. That is, the observational record is not necessarily representative of the distribution and likelihood of the uppermost ESL. The spread of the simulated $\mathrm{RL}_{100}$ from the 10 subsets is around twice as large as the 95% confidence interval of the estimated $\mathrm{RL}_{1000}$ using a Gumbel distribution fit onto the observed 100-year data (green bar). This uncertainty range of the Gumbel fit doubles though if the spread in $\mathrm{RL}_{100}$ is taken into account (grey bar). The non-parametrically obtained

$\mathrm{RL}_{1000}$ lies with 3.7 within both distribution ranges, but closer to the median of the Gumbel distribution fits. Considering the large variations and corresponding uncertainties it is obvious that high-impact return levels (or flood protection standards for that matter) cannot reliably be inferred from short time series. The sample size considered as a base for parametric extreme value analysis affects not only the likely range of high-impact return events, but may also lead to a problematic negligence of potential scenarios.

How the data record combined with the choice of extreme value distribution impacts the estimation of high-impact return levels is illustrated in Fig. 10: For this, we fit the GEV distribution and its special case of a Gumbel distribution to shorter subsets (33 x 30 year and 16 x 60 year segments, respectively) of the full 1000 annual maximum sea levels and compare the $\mathrm{RL}_{100}$ estimates to the non-parametrically obtained $\mathrm{RL}_{100}$ (Fig. 10). A doubling of the data length from 30 to 60 years, for instance, roughly reduces the range of the $\mathrm{RL}_{100}$ from each segment's distribution fit to half and the uncertainty range to $1/3$

(GEV fit) or $2/3$ (Gumbel fit), respectively. The flexible GEV distribution gives more weight to the strongly varying tails of the distribution and results in a larger range of estimated $\mathrm{RL}_{100}$, which is mostly due to the variations in the shape parameter $k$ which strongly depends on the respective subset. If $k$ is held constant (which is often assumed in non-stationary extreme value analysis, e.g. in Mudersbach and Jensen (2010)), the uncertainty range reduces considerably. In contrast, the Gumbel fit with a constant zero shape parameter leads to a narrower estimate of $\mathrm{RL}_{100}$. Yet, both distribution fits tend to favor the lower end of

simulated 100-yr return levels, although the $\mathrm{RL}_{100}$ inferred non-parametrically from the full simulation (green bar) lies within the inter-quartile range of both distributions' estimates. This discrepancy can also manifest itself in the temporal variations: The parametric return value estimates can further exhibit different temporal behavior in comparison to the non-parametric plotting positions of the full simulation, and may even lie outside the associated uncertainty range. That is, even such non-stationary extreme value analysis (e.g. Méndez et al. (2007); Mudersbach and Jensen (2010)) does not necessarily reflect the

'real' variations in high return levels.





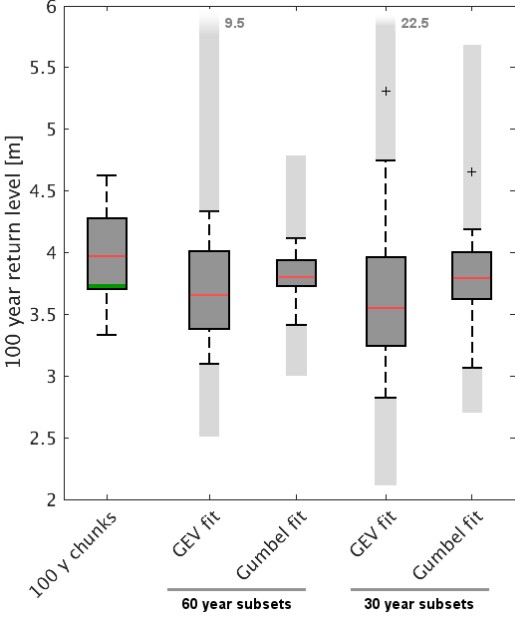

**Figure 10.** Box-Whisker-plots of $RL_{100}$ non-parametrically obtained from $10\times100$-year segments of the full simulation against $RL_{100}$ estimates based on GEV and Gumbel distributions fitted to 60-year and 30-year subsets using maximum likelihood. The range of the dashed whiskers represents the total range of the fits, light grey shading represents the maximum range of the 95% confidence limits of the respective fits. The red lines indicate the median of the realizations, the green lines mark the $RL_{100}$ directly inferred from the full 1000-year long simulation.

Thus, existing ESL estimates based on limited data lengths do therefore not reflect the full range of ESL variability, e.g. if data stems from a period of unusual state of a driving mechanism such as the SLP dipole outlined above.

Putting these findings into the context of climate change, mean sea level rise will – following a shift of the entire distribution

5    – accordingly translate into a higher probability of occurrence of a particular level and lower return periods. However, while the probability distribution shifts to the right, the large noise in the simulated ESL time series and the low explained variance by BSL imply that in the short term, German Bight ESL might in reality show a much stronger or even a reversed trend.

Yet, additional to a simple shift, a change in the shape of the distribution with increasing atmospheric GHG concentrations can further complicate the picture. Such a change may arise from changes in storm surge climate (e.g. due to an intensification

10    and poleward shift of storm activity (e.g., Yin, 2005; Fischer-Bruns et al., 2005)), or from a damping effect on the wind surge in a deeper sea, although this effect has been found to be comparably small (Lowe et al., 2001). This uncertainty in future storm surge projections is expressed by differing findings in existing studies that show a spectrum from no (e.g., Sterl et al., 2009) over little (e.g., Langenberg et al., 1999; Woth, 2005) to considerable change (Lowe and Gregory, 2005). As the variations of ESL are of one order of magnitude higher than the corresponding BSL ones (see Fig.3), the variance explained by the latter





is low and the detection of changes in the distribution difficult from a statistical point of view. With the strong, but random fluctuations of ESL on timescales of years to decades, we expect existing estimates of ESL changes to be dominated by natural variability rather than climate change signals. Large ensemble simulations will be necessary to detect any significant change in ESL statistics in the presence of the high natural variability found in our simulation. Sterl et al. (2009) have already addressed

this issue by using a large ensemble (17 members) of the SRES A1b climate change scenario run with a regional storm surge model and found no signal from climate change on $RL_{10\,000}$ along the Dutch coast. Yet, most studies evaluating future storm risks on data shorter than the estimated return periods, either from observations or from scenario simulations, do not account for such large ensembles and thus systematically disregard ESL variability on timescales longer than the baseline periods. The ESL variation can be substantial though as the large spread of simulated upper-end return levels during the last millennium

has shown. For instance, assuming a sea level rise of 0.5 m until the end of the century and given the here quantified ESL variability, more than roughly 200 (350) years of data would be necessary for $RL_{100}$ estimates (95% confidence bounds) using a Gumbel fit to range over less than the estimated signal. Without using large ensembles, ESL projections may be biased by the respective baseline period for extreme value analysis; and even worse, with a small ensemble size or one realization (e.g. the instrumental record) only we cannot say whether they are biased or not. On the other hand, the high internal variability

which is essentially irreducible (Fischer et al., 2013) also implies that even perfect models cannot provide well-constrained information on local ESL changes from one realization that might be desirable for adaptation planners.

A couple of caveats that may have an influence on our results are worth discussing. Simulated ESL are – relative to the long-term mean – too low, which is most likely related to an under-representation of the tidal range in the German Bight (see

Supplementary Fig. A2) and to simplifications in the model bathymetry: the model's minimum water depth of 16 meter exceeds the real depth of the shallow waters in many coastal areas of the German Bight and thus likely leads to lower wind surges than observed. Yet, in terms relative to mean high waters, simulated ESL statistics agree well with observations, and the temporal variability is not affected by this.

Moreover, the model does not allow for changes in bathymetry, shoreline or coastal management that may influence relative

sea levels. This may benefit the homogeneity of the simulated sea level variations, but may hamper the comparability to observations. Processes such as changes in local bathymetry, wave interference at ports or simply inconsistencies in the data record can obstruct the homogeneity of observations from tide gauge records. Additionally, discrepancies between the exact tide gauge position and the nearest-neighbor grid-box can further complicate the picture. A direct comparison between simulated and observed sea levels should therefore be treated with caution. Furthermore, a transient sea level rise due to the melting of

ice sheets is not accounted for in the model and a potential increase in ESL with a gradual rise in the BSL base could not be investigated, but will be addressed in a follow-up study.

Finally, the results were obtained by downscaling simulations from one GCM only. Potential biases in the parent GCM can thus feed into the downscaled results. For instance, both Northern Hemispheric storm tracks as well as the North Atlantic Current have been found to be too zonal in ECHAM and MPIOM, respectively (Sidorenko et al., 2015; Jungclaus et al., 2013).






As the downscaling of another ensemble member of the parent GCM simulations has shown, the temporal ESL variations of the two different downscalings differ significantly (see Supplementary Fig. A5), albeit their long-term statistics are comparable. That is, any external influence on long-term ESL variability is negligible and it is the natural variability of the parent GCM which determines the temporal variability on a larger scale. The regionalization, however, can offer more detailed dynamical patterns such as blockings. With more precise wind speeds and directions as well as the consideration of regional shelf dynamics it is thus more important for individual surge heights and finer regional differences. The variability due to the downscaling has been addressed by performing an additional downscaling of the same global simulation which has yielded similar spatio-temporal variability (see Supplementary Fig. A6), indicating that the temporal ESL variability directly linked to the downscaling is negligible. This is in accordance with Woth et al. (2006) who, comparing a number of regional climate models, concluded that the added uncertainty from the downscaling step of ESL variations from global to regional models was comparably small.

## 5 Summary and conclusions

Our study has provided the first coupled downscaling simulation focusing on storm surges and sea level evolution, which gives an unprecedented long high-resolution data record that can extend the knowledge of long-term ESL variability based on observations from tide gauge data which are limited in time and space. This simulation renders non-parametric extreme value analysis possible and has the advantage of not relying on extreme value distributions that are typically applied to short data series to provide information about return periods longer than the original time series, as well as their associated uncertainties. The special setup of coupling a high-resolution regional atmospheric model to a global ocean model including tides combines their respective advantages of (i) a consistent simulation of signals both inside and outside the region of interest, and (ii) a sufficiently high resolution in the region of interest to properly account for regional ocean-atmosphere dynamics and other shelf processes. At the same time, the continuous global simulation allows for setting the ESL variability into the context of simulated climate states.

The variability of extreme storm floods has been investigated through the means of annual maximum sea levels at Cuxhaven; the results obtained from different extreme value indices and other gridpoints along the German Bight coastline however do not differ significantly, suggesting that the qualitative behavior and variability are robust and do not depend on the extreme value sampling method or exact location.

Our results suggest that

1. The model reproduces observed storm surge statistics at Cuxhaven, both in terms of seasonality as well as magnitude above mean high waters.

2. ESL variations are large, but operate on a white spectrum and do not exhibit significant oscillatory modes beyond the seasonal cycle.



3. High-impact extreme events vary substantially on timescales longer than the typically available base period for return period estimates. Estimates of ESL obtained via the standard parametric approach based on short data records are therefore not representative for the full ESL variations.

4. Long-term ESL variations in the German Bight are regionally consistent, indicating a common large-scale forcing. Large-scale circulation regimes that favor periods of enhanced ESL in the German Bight are similar to those associated with elevated BSL, but the location of the respective centers of action of the governing SLP dipole differs. While BSL variations correlate well with the wintertime NAO, ESL variations are rather associated with a shifted NAO+/SCA- like pressure pattern leading to a stronger local northwesterly wind component.

5. Any potential links to BSL fluctuations as well as external influence through solar variability or volcanic activity are masked by the strong internal variability of ESL. This is in accordance with the findings by (Fischer-Bruns et al., 2005) who, using coupled *Last Millennium* simulations of the last five centuries, concluded that the natural variability of mid-latitude storms is not related to solar, volcanic or GHG forcing nor to anomalous climate states such as the Maunder Minimum. Similar conclusions have been made for North Atlantic summer storm tracks over Europe by combining observations, simulations and reconstructions of the last millennium (Gagen et al., 2016).

We thus conclude that the magnitude of ESL and existing estimates of changes thereof are dominated by natural variability rather than forced signals. Given the large variability from our simulation, large ensemble simulations are required to detect a potential future change in ESL statistics with respect to climate change induced BSL.

Nevertheless, the obtained information on the statistics of ESL variability together with the here established links to large scale climate variability may be used to better explore future pathways of extreme sea levels. Owing to the large ESL variability, a responsible adaptation strategy should therefore reflect the range of possible developments rather than solely being designed to a forced signal. At the end, uncertainties in both SLR projections as well as ESL estimates need to be better understood and combined to fully assess potential impacts and required adaptation measures.

*Code and data availability.* Primary data and scripts used in the analysis that may be useful in reproducing the work are archived by the Max Planck Institute for Meteorology and can be obtained by contacting publications@mpimet.mpg.de.



# Appendix: Supplementary information

## A1   The instrumental record

Observations stem from the tide gauge record at Cuxhaven (courtesy S. Dangendorf & W. Wiechmann; see Fig. A1). Comparison of simulated and observed extreme sea level at Cuxhaven shows that the tidal amplitude is underrepresented (Fig. A2).

5   However, relative to the long-term tidal mean high water, seasonal statistics of storm surges agree well (Fig. A3), while daily weather situations are found to behave similarly to observations (not shown).

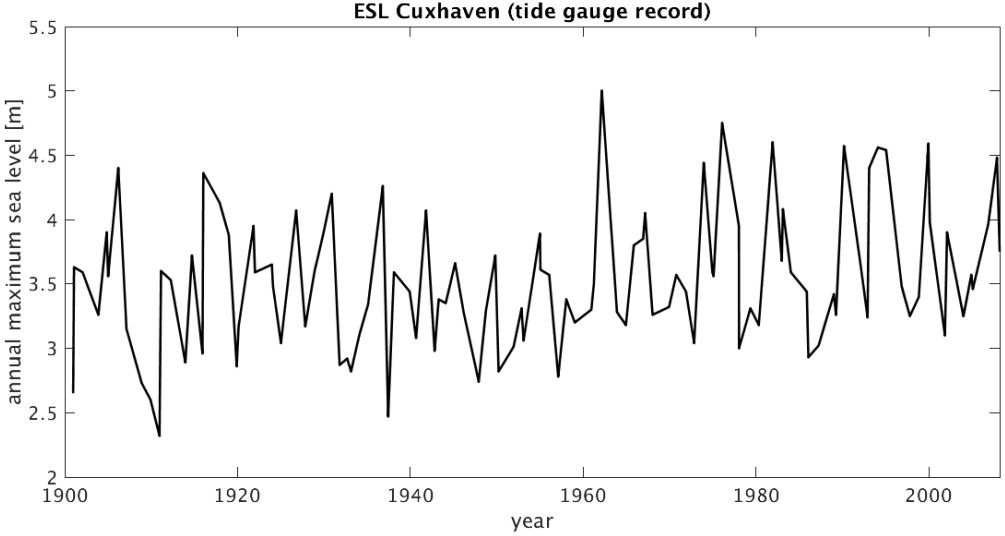

**Figure A1.** ESL in terms of annual maximum sea level from the tide gauge record at Cuxhaven. Values are given in meter above the long-term mean of the original data set of hourly resolution.

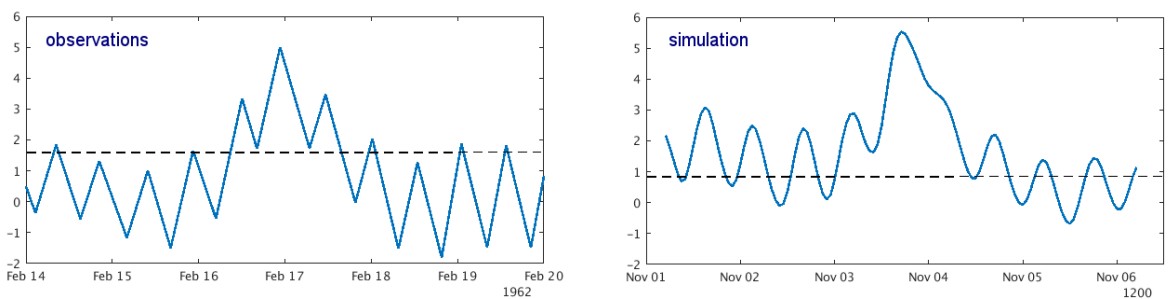

**Figure A2.** Time series of highest storm surge from observations (left) and model simulation (right). The dashed black line indicates the long-term mean high water. The respective long-term mean has been removed for both time series.





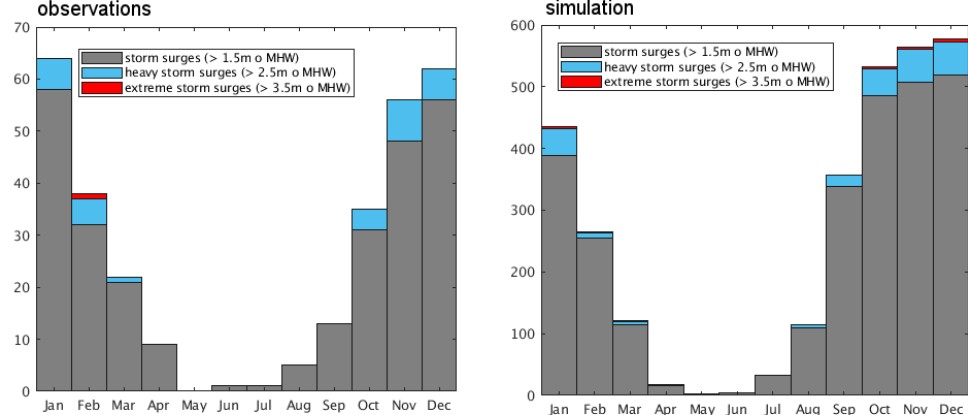

**Figure A3.** Storm surge frequency per month for observations (left) and model simulation (right) for storm surge classes following the definitions by the Federal Maritime and Hydrographic Agency (BSH).



## A2    External forcing for simulations

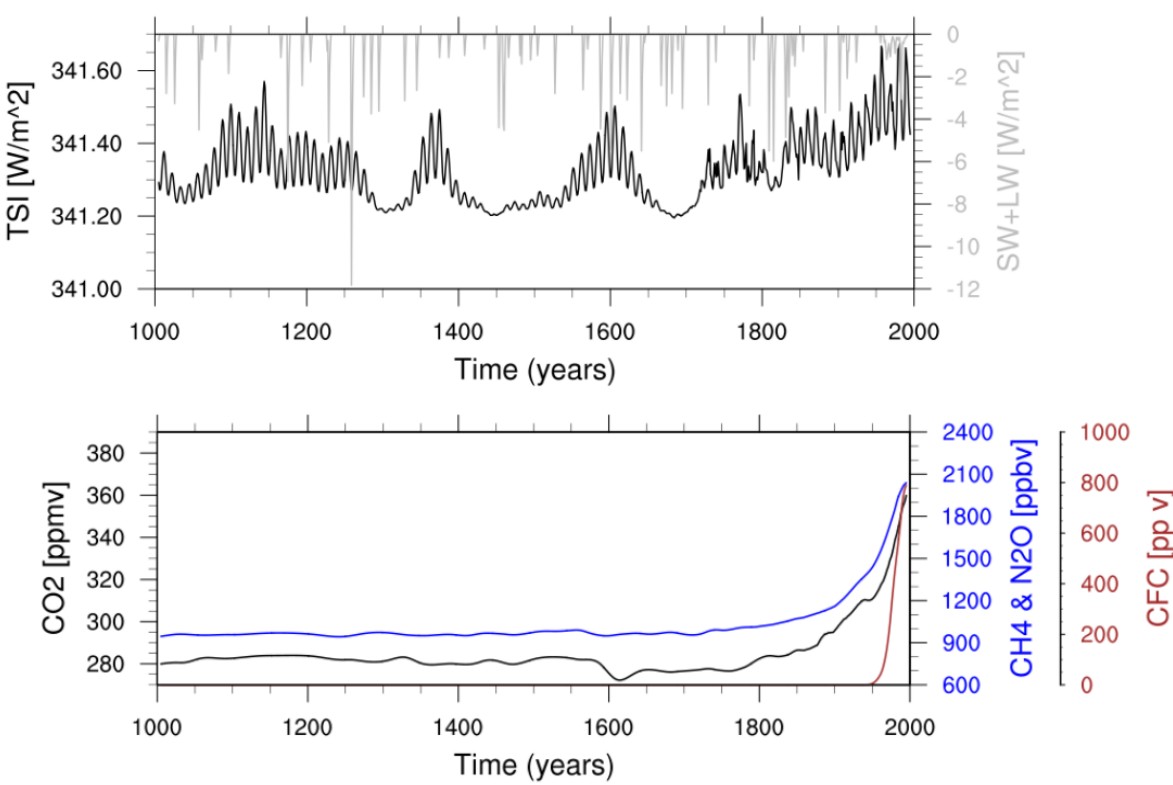

**Figure A4.** *Top:* Forcing input of annual solar irradiance [Wang et al. (2005), black line] and volcanic eruptions [Crowley et al. (2008), grey line]. *Bottom:* Greenhouse gas forcing, including $CO_2$ (black), $CH_4$ and $N_2O$ (blue) and Chlorofluorocarbons (CFC, dark red).





## A3 Downscaling simulations and sources of variability

In order to investigate the contributions of external forcing and natural variability on ESL variations, we have – additional to the downscaling shown above (experiment *011*) – downscaled a second member of the parent global *Last-Millennium* simulations ('past1000r1', see Moreno-Chamarro et al. (2017b)), covering the period 1400-1850 (experiment *012*). The first 100 years have been used as spin-up. While the ESL statistics in terms of a quantile-quantile plot are comparable, the two downscalings show different temporal ESL variations, indicating that the externally forced variability is small compared to the natural variability.

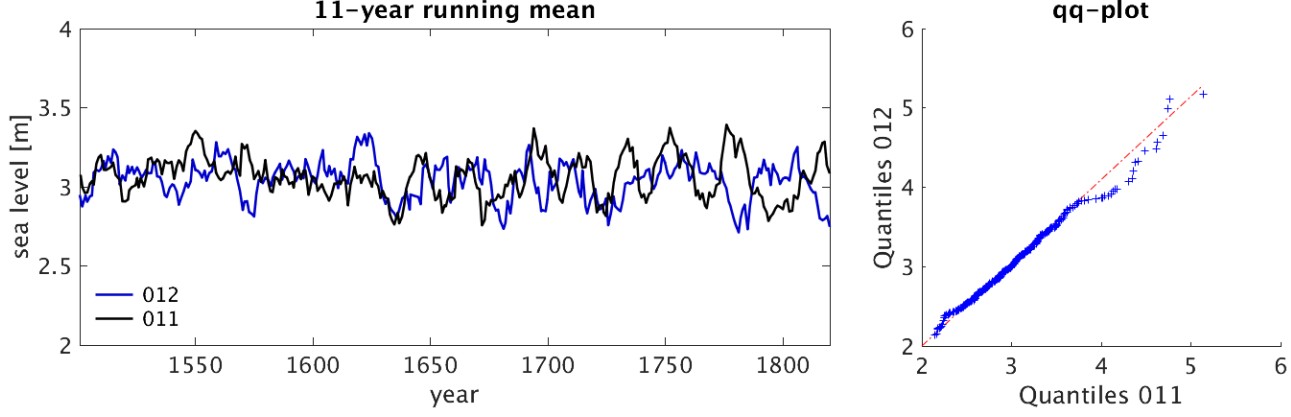

**Figure A5.** Comparison of ESL at Cuxhaven between runs *011* and *012* for the years 1500-1850. Left: Time series for 11-year running means. Right: Quantile-Quantile plot.

In order to investigate the effect of the downscaling process on ESL variability, the global PMIP3 simulation 'past1000r2' (1000-1850, (Moreno-Chamarro et al., 2017b)) and the subsequent 'historicalr4' (1850-2000) have been downscaled twice, once over one continuous 1000 year simulation (1000-2000AD.; experiment *011*; used for results shown in the main text), and once over the second 500 years (1500-2000AD.; experiment *010*) with slightly different initial conditions. Both simulation are preceded by a 100 year long spin-up. An overview of the downscaling simulations is given in table A1.

Although there is a tendency towards higher extremes in 010, especially for return periods greater than 30 years, the main features of long-term variability and spectral characteristics are not affected by the downscaling (Fig. A6). Nonetheless, the downscaling with REMO-MPIOM allows a more detailed simulation of large-scale dynamics (e.g. European blocking events).

**Table A1.** Downscaling simulations used in this study

| run ID | parent GCM simulation (MPI-ESM) | time period |
| --- | --- | --- |
| *010* | past1000r2 + historicalr4 | 1500-2000 (+100 year spin-up) |
| *011* | past1000r2 + historicalr4 | 1000-2000 (+100 year spin-up) |
| *012* | past1000r1 | 1500-1850 (+100 year spin-up) |





**Figure A6.** Comparison of ESL at Cuxhaven between runs *010* and *011* for the years 1500-2000. Left: Time series for 11-year running means. Right: Quantile-Quantile plot. Bottom: Return value plot of simulated sea level at Cuxhaven [m over MHW] (colored lines representing 100-year long segments of the full 1000 years) against observations from tide gauges (green crosses) for run *011* (red, 1000-2000) and for run *010* (blue, 1500-2000)





## A4  ESL Variability: Spatial coherence

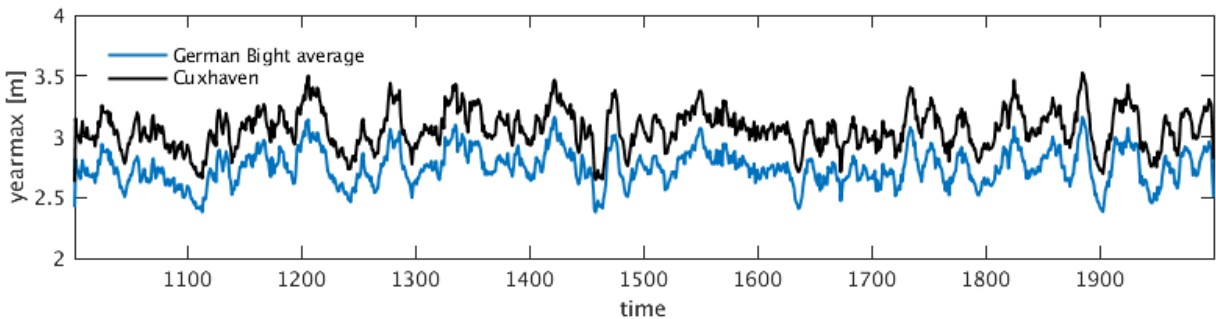

**Figure A7.** 11-year running means of the annual maximum sea level at Cuxhaven (black) and the spatially aggregated annual maxima along the German Bight coast (41 grid points, blue)





## A5 ESL Variability: Relation to BSL

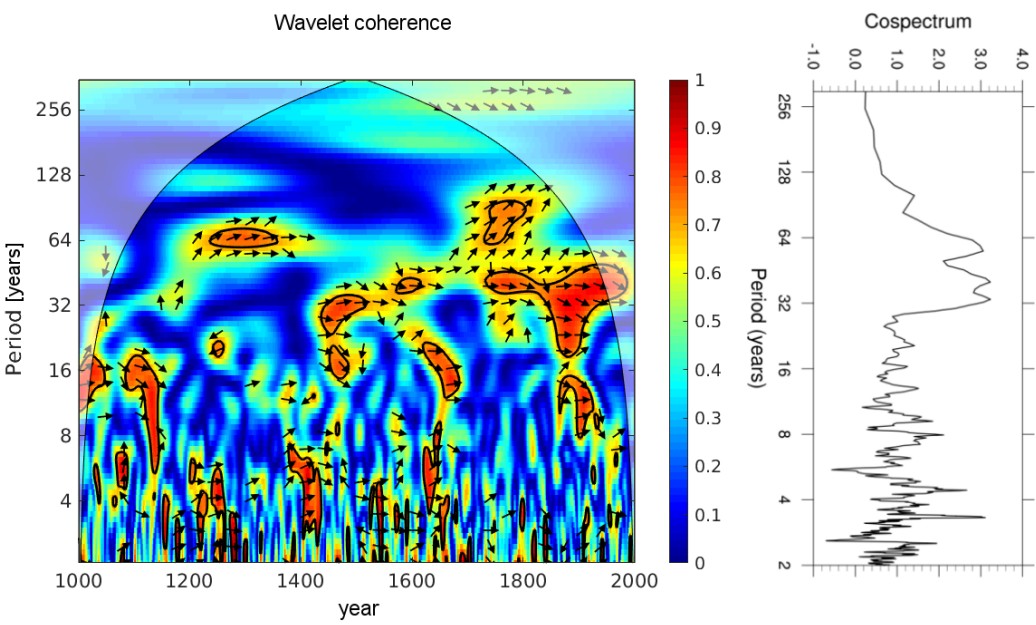

**Figure A8.** Wavelet coherence and cospectrum of winter median and annual maximum sea level. Arrows to the right (left) indicate a positive (negative) correlation and upward (downward) arrows indicate a lag (lead) of the first time series. Thick contours designate the 5% significance level against red noise, the cone of influence is shown as a lighter shade.

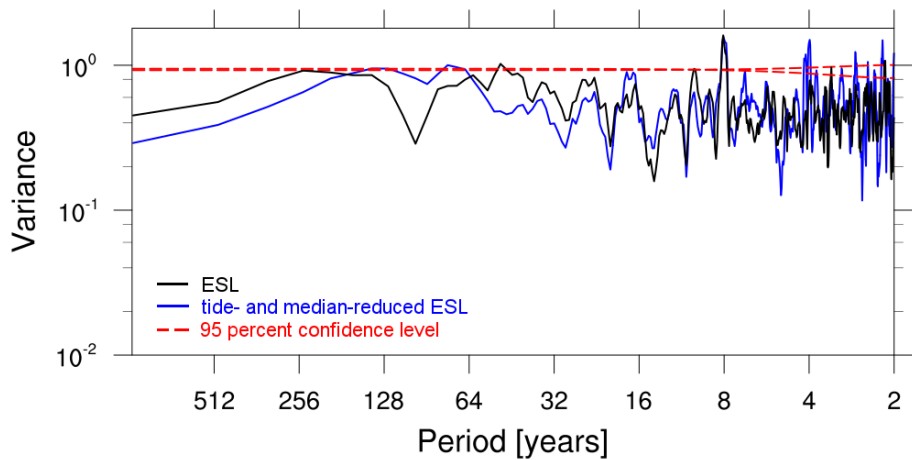

**Figure A9.** Spectra of ESL (black) and tide-and-median-reduced ESL (blue)





## A6 ESL Variability: Relation to external forcing

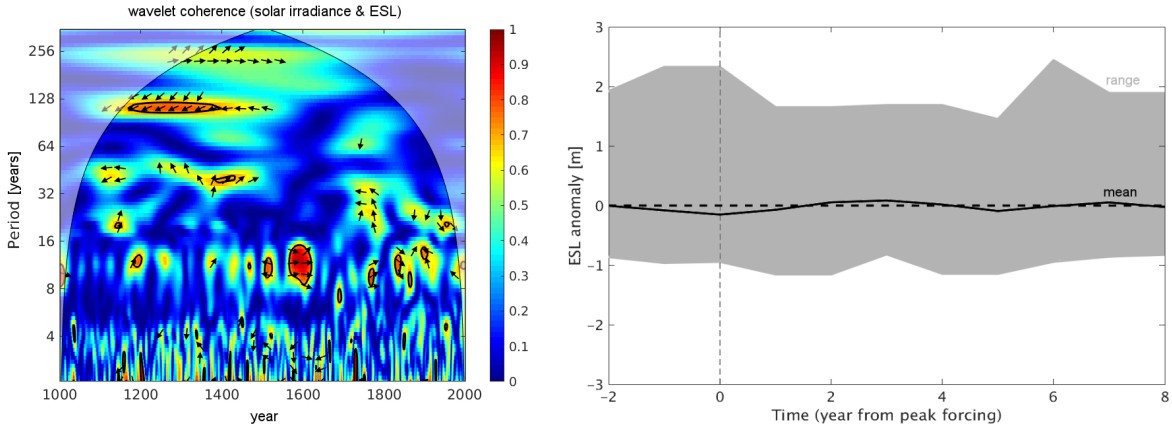

**Figure A10.** *Left:* Wavelet coherence between solar irradiance and ESL. *Right:* Mean lagged ESL after major volcanic activity (perturbance $> 4\,\mathrm{W\,m^{-2}}$).

*Author contributions.* UM came up with the idea for the manuscript and developed the model code. AL and UM jointly designed the experiments; simulation and analysis were carried out by AL. The manuscript was prepared by AL with contributions from UM.

*Competing interests.* The authors declare no competing interests

5    *Acknowledgements.* The research was supported by the Max Planck Society and by the German Research Foundation (DFG) funded project SEASTORM under the umbrella of the Priority Programme SPP-1889 'Regional Sea Level Change and Society' (Grant No. MI 603/5-1). The authors acknowledge Eduardo Moreno-Chamorro and Johann Jungclaus for producing the *past1000* global simulations and Sönke Dangendorf and Wilfried Wiechmann for providing the observational data from selected tide gauges. We thank Johann Jungclaus and Moritz Mathis for their constructive comments and remarks which helped to improve this manuscript. We further acknowledge the German Climate
10   Computing Center (DKRZ) for providing the necessary computational resources.





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
