# Peer review of "The long-term variability of extreme sea levels in the German Bight"

_Ocean Science, 2019_

## Referee Comment (RC1) · Anonymous Referee #1 · 27 Mar 2019

The manuscript presents an interesting study on the long-term evolution of extreme sea-levels in the North Sea Sea, as simulated by a rather unique model chain including a global ocean model and regional atmospheric and ocean models. The period of analysis covers the past millennium. The study focuses on the the connections between the variations in extreme sea-level and the mean sea-level, their causes of variability (external forcing, internal climate variability), and the uncertainties in the estimation of return value statistics. The main conclusions are (1) that the variations in the statistics of extreme sea-level are mainly driven by internal climate variations; (2) that these variations are only weakly connected to the variations of background sea-level; (3) that the estimation of return values based on short observational records are hampered by larger uncertainties as assumed so far.

[Figure]

My impression of the manuscript is quite positive. The study is also relevant and I am happy to recommend it for publication. I have a few suggestions to some particular points in the manuscript, but in general my opinion is that the manuscript is in quite good shape. Some of my suggestions refer to the English formulation- these should be doubled checked by the authors

1. 'Extreme sea levels particularly arise when these components are in superimposition'

superposition

2. 'Yet, a comparison in terms of extreme value statistics is possible. considering storm flood statistics, we compare the simulated ESL with observations from..'

I think the authors mean 'meaningful' rather than possible. A comparison is always possible, but it may be conceptually wrong

3. The return values at Cuxhaven derived from observations seem to be biased low. The authors write '...rather underrepresented, pointing to a bias towards too zonal (westerly) winds'. I am not sure whether this points to a bias. The estimations from the simulated 100-year segments cover the observations-derived value. As we only have one observations-derived value we cannot assert , I think, that the (theoretical) distribution of observational values is biased relative to the distribution of modelled values. I would rather write that the observational value is at the lower x-quantile of the model-derived distribution.

4. Figure 4 includes a label 'observed'. However, these time series are not observed per se, they are derived from observations, and these derivations can be obtained with different estimation methods, e.g. POT or GEV. I would use 'observations-based' or 'derived from observations'.

5. 'In agreement with observational studies (Gerber et al., 2016), simulated storm floods at Cuxhaven stem from predominantly west-north-westerly directions, while their

associated daily pressure anomaly patterns are similar to observations of storm flood weather situations (Donat et al., 2010; Dangendorf et al., 2014c)'

It would be helpful for the reader to include here either a reference to Figure 8 or, if the authors do not wish to disturb the ordering of in-text figure citations, to mention that this SLP pattern will be shown later. Otherwise, the reader may get stuck here wondering if the paper will show it or not.

6. The correlation between BSL and ESL is comparably low (r = 0.35) and highly variable over time (see black curve in Fig. 6 for a 100 year running correlation), while the different magnitudes of variances lead to a low explained variance

Here, it would be interesting to show the correlation between the indices describing the intensity of the SLP patterns that are linked to BSL and ESL (shown in Figure 8), and maybe also mention the correlation of both to the NAO index. This , it would become even more clear that, as I expect, the SLP pattern behind ESL is indeed different from the NAO, which in itself would be (the confirmation of) a quite important result.

7. '....trends between ESL and BSL during the last century that have often been described (Kauker and Langenberg, 2000; Menéndez and Woodworth, 2010) might merely be an unusual state if compared to a longer time horizon as obtained from our long-term simulation.'

This is perhaps my more substantial comment. The 20th century observations, according to this paragraph, show that there is a link between background sea-level and extreme sea-level in that century. It is not clear to me whether the model run also shows this link, and it also not clear to me which mechanisms may explain this link in reality. Is it that the two patterns in Figure 8 have tended to evolve coherently in the 20th century and not in previous centuries ? is it that the strong background sea-level rise in the 20th century has affected the probability of ESL, and that in previous centuries the variations in BSL were not strong enough to supersede the influence of internal atmospheric variability? In this regard, an important question is whether or not the

simulation is able to replicate the observed sea-level rise both at global and at regional scales due to thermal expansion and maybe also to forced changes in the large-scale ocean circulation (AMOC), Perhaps I missed it in the manuscript, but I think this information is not present, and I think it is relevant, since it would support the simulation of past BSL variability. At least the thermal expansion component in the model and in the observations since 1950 should roughly agree at global scales. In Figure 3 (green line) it is difficult to eyeball. It seems that the regional not-land-ice related sea-level rise in the 20th century is not remarkably different from past variability, but accurate numbers would help the reader.

8. Fig 6 . Time series have been smoothed with a 11y moving window.8.

It is not totally clear to me how this has been calculated. Were the 100-year gliding correlations later smoothed with a 11-year running mean, or where the initial series first smoothed and then the 100-year gliding correlations calculated. This is important for the set-up of the bootstrapping.

9. range of the Gumbel fit doubles though if the spread in RL100 is taken into account (grey bar). The non-parametrically obtained RL1000 lies with 3.7 within both distribution ranges, but closer to the median of the Gumbel distribution fits. Considering the

3.7 meters, I guess
* * *

---

## Referee Comment (RC2) · Anonymous Referee #2 · 28 Mar 2019

General Comments: The manuscripts investigates the sea level variability in the German Bight on millennial time scales, using a downscaling technique from a global GCM. Several important and interesting questions are tackled regarding for example: trend detection, connections between atmospheric patterns and sea level extremes and the fidelity of extreme value statistics derived from the operational records.

The manuscript is most well written, and interesting. I believe it will make a nice contribution to the journal after some minor revisions.

Specific comments: 1: My biggest objection to the manuscript is the structuring, where I think the supplementary material is much to extensive. Parts of it like the comparison to observations would be better put in the text, and other parts, I think, could be dispensed of. Specifically I think figs. A2,A3,A7 would be fit better as part of the article.

[Figure]

Figs A1,A9 and A10 I believe did not add much and could likely be dispensed of. However, I think if would be nice to add quantile plots of model vs observations for different 100 year periods, too have a more direct comparison to observations

2:The 16 m minimum depth. This is discussed a bit, but not in relation to the representation of bottom topography. Being myself unfamiliar with the German Bight, I don't know if representation of complex bottom topography is a problem in the area. However, I think that it should be stated in the article if this is a potential problem or if the bottom is essentially flat.

3: Sect. 3 how mean high water is defined should be in the paper.

4: Sect. 3.1 row 25 comparison to observations. It is stated that ESL relative to MHW compares well to observations, but to me it seems that ESL is likely somewhat overestimated. I think the suggested quantile plots would be helpful here.

5: Fig 4. I was first confused over how the return levels were estimated here. I think it should be stated in the caption that they are non-parametric, and I also suggest that you add a second panel showing there parametric counterparts for the same time period. Also a non-parametric return level curve for the whole data set would be interesting.

6: Page 10 just before Sect. 3.2, it is stated that other grid points are similar to Cuxhaven. I don't dispute this, but the figure shows a mean of the other gridpoints, and says noting about the spread, so I would rephrase.

7: Sect. 3.2 What about the correlation of the running mean series from Fig 4. It looks like it might be higher?

8: Fig.9 and all wavelet figures. Please state which series leads which for a given arrow direction, not just the first series.

9: Sect. 3.4. I think you could develop the last thoughts in this paragraph more, and also show the SLP pattern when BSL has been removed as an additional panel in Fig. 8

10: Page 19 around line 30. You mention the lack of melting ice-sheets, but not the thermosteric effect. I assume your model is Boussinesq so you don't have this either?

Technical corrections: 1: End of page 1 start off page 2, you could get rid of the German Bight at the end.

2: Page 7 line 3-4. This sentence seems misplaced in the text.

3: Page 7 line 30. Yet, other ..... –> However,?

4: Page 13 line 13-16, some odd sentences here, missing a .

5: Page 17 line 28, In contrast –> Similarly?

6: Page 17 line 29-35, The sentence is difficult to read

7: Page 19 line 14. Only?

8: Fig A3 I would prefer a normalised y-axis

---

## Referee Comment (RC3) · Anonymous Referee #3 · 11 Apr 2019

formal assessment

Does the paper address relevant scientific questions within the scope of OS? yes

Does the paper present novel concepts, ideas, tools, or data? yes

Are substantial conclusions reached? yes

Are the scientific methods and assumptions valid and clearly outlined? yes

Are the results sufficient to support the interpretations and conclusions? yes

Is the description of experiments and calculations sufficiently complete and precise to allow their reproduction by fellow scientists (traceability of results)? yes

[Figure]

Do the authors give proper credit to related work and clearly indicate their own new/original contribution? yes

Does the title clearly reflect the contents of the paper? yes

Does the abstract provide a concise and complete summary? yes

Is the overall presentation well structured and clear? yes

Is the language fluent and precise? yes

Are mathematical formulae, symbols, abbreviations, and units correctly defined and used? yes

Should any parts of the paper (text, formulae, figures, tables) be clarified, reduced, combined, or eliminated? yes

Are the number and quality of references appropriate? yes

Is the amount and quality of supplementary material appropriate? yes

general comments

The article "The long-term variability of extreme sea levels in the German Bight" by A. Lang and U. Mikolajewicz addresses an urgent and important issue in the current discussion on extreme sea levels (ESL) and how they might evolve in a changing climate. Using a millennial long, downscaled coupled model run of the past 1000 years they show by example of the Cuxhaven tide gauge how standard approaches of ESL estimation based on decadal to centennial long time series underestimate the uncertainty inherent in the natural variability of ESL. They show that the ESL variability exhibits a white spectrum and that the atmospheric pattern related to high ESL resembles the negative phase of the Scandinavian pattern. By comparing their time series with an additional one from a model run that downscaled a run with different initial conditions they show that the trajectories are different and the statistics comparable, which implies that external forcing (e.g. solar or volcanic) is not a major driver of ESL variability. The

large uncertainty in ESL from natural variability should have an impact on how future ESL estimates based on observational time series are interpreted.

The article is innovative, relevant and well written and should be published in OS after some minor issues have been addressed.

specific comments

- The abstract is good and might be even easier to grasp for someone not working in the field if not all abbreviations are introduced there. The NAO or SLP could be introduced later, since they are mentioned only once in the abstract.

- Could you mention briefly how those processes that are not included in the model, like ice sheet melt or land uplift might alter your ESL analysis.

- At the end of the first paragraph in Section 3.1 you mention that the long-term trend from the observations was removed. Was it a linear trend and was the residual of the fit tested for being mostly white noise?

- In the caption of Figure A2 you mention that "The respective long-term mean has been removed for both time series." and you show the MHW as a dashed line in the figure. The information from the figure and the caption might become clearer if either both the long-term mean and the MHW would be indicated in the figure or was it anyway the MHW which was removed as implied in the text in Section 3.1?

- I am not against a conservative approach to show only the 50-year return values from a 50 year long record. Others calculate up to three-fold return periods given the length of a time series. Are there reasons to not calculate the 100-year return level at Husum and Norderney?

- In the last paragraph of Section 3.1 you argue that the model is spatially coherent along the German Bight coast and in Figure A7 you show the average along the coast. I think you need to show the median with interquartile range or something similar to support that claim. It would allow the reader to assure herself that local effects and

such are not playing an important role.

- On Page 12 you argue that the pattern of the composites are robust against minor changes in the threshold of what constitutes "high ESL". It might help the reader to integrate "minor changes" into the story a little better if those changes could be related to a number. For example mean plus 1.5 standard deviations plus/minus 0.25 standard deviations, or 80 to 120 periods of "high ESL".

- In the discussion on page 17 the non-parametric estimates of return levels are compared to various estimates from fitting distributions with certain parameters to the sampled distributions. I think this discussion would become clearer if the non-parametric estimates had been introduced in a sentence or two. How did you exactly determine the non-parametric estimates? This could also be added to the method part.

- On page 19, line 29 you mention that the GCM does not account for melting ice sheets. Is the global mean thermosteric effect included in the analyzed sea level?

technical corrections

title: Uwe Mikolajewicz

p5l10: global Earth System Model MPI-ESM

p5l18: Topography and coastlines as well as ice sheets are immutable

p6l3: with the first 100 years again used as spin-up

p6l24: fitted to the comparably short data series.

p7l30: the return values inferred from observations lie within the spread of the model simulation at all periods.

p8l3: [I am not sure myself whether "section 4" is a name and should be capitalized?]

p13l9: This pattern is different from the meridional NAO-like dipole

Figure 9: Left: Pointwise regression of the SLP index derived from Fig. 8 onto annual

[Figure]

maximum sea level

p15l23: plementary Fig. A5) stresses the dominance

p17l15: The non-parametrically obtained RL1000 lies with 3.7 m within both distribution ranges

Figure 10: the green line mark the RL100

Figure A2: The respective long-term mean has been removed from both time series.

p25l11: [Same issue as above: table A1 is probably a name that should probably be capitalized.]

---

## Author Comment (AC1) · 15 May 2019

**The long-term variability of extreme sea levels in the German Bight**

**Andreas Lang and Uwe Mikolajewicz**

**Authors' response**

**RC1 Review by Anonymous Referee #1**

We thank Anonymous Referee #1 for the helpful comments. The individual comments are addressed below. Page and line numbers in the responses refer to the updated version of the manuscript; changes therein are marked in red.

1. Comment by Referee:
   *'Extreme sea levels particularly arise when these components are in superimposition'*
   *Superposition.*

   Response:
   superimposition has been changed to superposition (p.2, line 23)

2. Comment by Referee:
   *'Yet, a comparison in terms of extreme value statistics is possible. considering storm flood statistics, we compare the simulated ESL with observations from..'*
   *I think the authors mean 'meaningful' rather than possible. A comparison is always possible, but it may be conceptually wrong.*

   Response:
   The corresponding sentence has been clarified (p.7, line 26f.)

3. Comment by Referee:
   *The return values at Cuxhaven derived from observations seem to be biased low. The authors write '...rather underrepresented, pointing to a bias towards too zonal (westerly) winds'.*
   *I am not sure whether this points to a bias. The estimations from the simulated 100-year segments cover the observations-derived value. As we only have one observations-derived value we cannot assert , I think, that the (theoretical) distribution of observational values is biased relative to the distribution of modelled values. I would rather write that the observational value is at the lower x-quantile of the model-derived distribution.*

   Response:
   We do not claim that the return values at Cuxhaven are biased low, they indeed lie within the simulated spread. However, the sentence the referee is referring to concerns sites along the coastline of Lower Saxony (see Fig. 6 (former Fig. 5)), where the simulated return levels are – other than at Cuxhaven – *lower* than the observation-based return levels. We agree though that the word 'bias' is a bit far fetched as we indeed do not know the theoretical distribution of observational values based on one single observation-derived time series. Therefore, the statement about a possible bias has been rephrased (p. 9, line 5f.):
   *'Yet, while the return values at Cuxhaven lie slightly higher than the observed ones, ESL along the coastline of Lower Saxony and the Netherlands are rather low compared to the observation-based estimates.'*

4.  Comment by Referee:
    *Figure 4 includes a label 'observed'. However, these time series are not observed per se, they are derived from observations, and these derivations can be obtained with different estimation methods, e.g. POT or GEV. I would use 'observations-based' or 'derived from observations'.*

    Response:
    The label has been changed accordingly. Note that due to a newly added Fig. 4, the mentioned Figure is now Fig. 5.

5.  Comment by Referee:
    *'In agreement with observational studies (Gerber et al., 2016), simulated storm floods at Cuxhaven stem from predominantly west-north-westerly directions, while their associated daily pressure anomaly patterns are similar to observations of storm flood weather situations (Donat et al., 2010; Dangendorf et al., 2014c)'.*
    *It would be helpful for the reader to include here either a reference to Figure 8 or, if the authors do not wish to disturb the ordering of in-text figure citations, to mention that this SLP pattern will be shown later. Otherwise, the reader may get stuck here wondering if the paper will show it or not.*

    Response:
    Fig. 8 (now Fig. 9) does not show a daily pressure anomaly pattern, but rather a composite for periods of enhanced ESL based on the 3-year low-pass filtered ESL time-series (see p.13, line 30). A direct reference to the Figure would be misleading at this point.
    We decided against showing a corresponding figure about the *daily* anomaly pattern for length and readability reasons. However, an explicit statement of this has been added (p. 9, line 14).

6.  Comment by Referee:
    *'The correlation between BSL and ESL is comparably low (r = 0.35) and highly variable over time (see black curve in Fig. 6 for a 100 year running correlation), while the different magnitudes of variances lead to a low explained variance.'*
    *Here, it would be interesting to show the correlation between the indices describing the intensity of the SLP patterns that are linked to BSL and ESL (shown in Figure 8), and maybe also mention the correlation of both to the NAO index. Thus, it would become even more clear that, as I expect, the SLP pattern behind ESL is indeed different from the NAO, which in itself would be (the confirmation of) a quite important result.*

    Response:
    A couple of sentences about the correlation between the SLP patterns behind ESL and BSL with the NAO have been added (p. 14, line 25f.): *"Compared to the pattern associated with high BSL (correlation with NAO of r = 0.9), this SLP pattern has a lower correlation with the NAO (r = 0.67). As a result, the ESL time series at Cuxhaven as well shows a weaker correlation with the NAO (r = 0.19) than with the newly defined SLP pattern (r = 0.31)."*

7.  Comment by Referee:
    *'....trends between ESL and BSL during the last century that have often been described (Kauker and Langenberg, 2000; Menéndez and Woodworth, 2010) might merely be an unusual state if compared to a longer time horizon as obtained from our long-term simulation.'*
    *This is perhaps my more substantial comment. The 20th century observations, according to this paragraph, show that there is a link between background sea-level and extreme sea-level in that century. It is not clear to me whether the model run also shows this link, and it*

*is also not clear to me which mechanisms may explain this link in reality. Is it that the two patterns in Figure 8 have tended to evolve coherently in the 20th century and not in previous centuries ? is it that the strong background sea-level rise in the 20th century has affected the probability of ESL, and that in previous centuries the variations in BSL were not strong enough to supersede the influence of internal atmospheric variability? In this regard, an important question is whether or not the simulation is able to replicate the observed sea-level rise both at global and at regional scales due to thermal expansion and maybe also to forced changes in the large-scale ocean circulation (AMOC), Perhaps I missed it in the manuscript, but I think this information is not present, and I think it is relevant, since it would support the simulation of past BSL variability. At least the thermal expansion component in the model and in the observations since 1950 should roughly agree at global scales. In Figure 3 (green line) it is difficult to eyeball. It seems that the regional not-land-ice related sea-level rise in the 20th century is not remarkably different from past variability, but accurate numbers would help the reader.*

Response:

There is currently no overall consensus on the link between background sea level and extreme sea level, as for instance Mudersbach et al. (2013) found differences in linear trends in high sea level percentiles from those in mean sea level. Yet, most studies report similar trends.

According to the running correlation in Fig. 7 (former Fig. 6) and the wavelet coherency spectrum in Fig. A7 (former Fig. A8), the stronger coherence bewteen ESL and BSL at multidecadal timescales in the last century is also a feature in our simulation. However, the high internal variability of ESL, which is independent of external forcing or BSL variations if longer times are considered, masks this coherent behavior. It is only apparent in some centuries, like the 20[th]. The reason for this is not clear. It is indeed possible that this is related to the associated large-scale circulation regimes: ESL and BSL related SLP patterns evolve rather similarly (corr = 0.86, vs 0.78 over whole period) in the 20[th] century; it might also be by chance though. However, it does not stem from a gradual climate-change induced background sea level rise, as – other than in the observations – this transient thermosteric effect is not accounted for in the model simulation (see also Comment 10 by RV#2).

We added a statement about the coherency of the two SLP patterns (page 14, line 27f.). Furthermore, we specified the omission of the thermosteric effect in the ESL analysis in the method and discussion sections.

Concerning the question whether the simulation is able to replicate the observed sea level rise due to changes in the large-scale ocean circulation: The model is generally able to simulate realistic sea level changes due to changes of the AMOC strength. A stronger overturning circulation, for instance, leads to a stronger SPG and reduced sea level in the region (see scatter-plot below). However, this response is less pronounced on the North West European Shelf.

[Figure]

[Figure]

[Figure]

8.  Comment by Referee:
    *Fig 6 . Time series have been smoothed with a 11y moving window.*
    *It is not totally clear to me how this has been calculated. Were the 100-year gliding*
    *correlations later smoothed with a 11-year running mean, or where the initial series first*
    *smoothed and then the 100-year gliding correlations calculated. This is important for the*
    *set-up of the bootstrapping.*

    Response:
    The initial series were first smoothed and then the 100-year gliding correlations calculated.
    A clarifying statement has been added (p.12, caption of Fig.7 (former Fig. 6))

9.  Comment by Referee:
    *… range of the Gumbel fit doubles though if the spread in RL100 is taken into account*
    *(grey bar). The non-parametrically obtained RL1000 lies with 3.7 within both distribution*
    *ranges, but closer to the median of the Gumbel distribution fits.*
    *3.7 meters, I guess*

    Response:
    Yes. The unit has been added. (p. 18, line 22)

All technical corrections have been incorporated.

---

## Author Comment (AC2) · 15 May 2019

**The long-term variability of extreme sea levels in the German Bight**

**Andreas Lang and Uwe Mikolajewicz**

**Authors' response**

**RC3 Review by Anonymous Referee #3**

We thank Anonymous Referee #3 for the helpful comments. Responses to the individual comments are listed below. Page and line numbers in the responses refer to the updated version of the manuscript; changes therein are marked in red.

1. Comment by Referee:
   *The abstract is good and might be even easier to grasp for someone not working in the field if not all abbreviations are introduced there. The NAO or SLP could be introduced later, since they are mentioned only once in the abstract.*

   Response:
   In order to sharpen the main message, the abstract has been rewritten. Abbreviations not needed have been left out.

2. Comment by Referee:
   *Could you mention briefly how those processes that are not included in the model, like ice sheet melt or land uplift might alter your ESL analysis.*

   Response:
   These processes affect global and regional sea level gradually, and can both add upon another or cancel each other out. Such processes act on long time scales, and thus have a transient and different effect rather than sea level variations on interannual to multidecadal timescales, which are addressed in this study. Impacts of such long-term processes are expected for the entire sea level distribution, as the mean state gradually shifts and, potentially, the distribution's shape also changes. The ESL analysis might hence be affected by gradual changes in the background sea level. Exclusion of these long-term processes, however, makes it possible to isolate for changes in ESL variability due to both dynamic and thermodynamic effects without the influence of a gradual change in BSL.
   A paragraph addressing these issues is already part of the discussion. However, a couple of explanatory lines have been added (p. 21, line 4f.):
   *'Furthermore, a transient sea level rise due to melting of ice sheets, post-glacial isostatic rebound or the thermosteric effect is not accounted for in the model and a potential increase in ESL with a gradual rise in the BSL could not be investigated. Such transient sea level changes can further impact ESL on longer time scales, since the sea level distribution shifts with changes in BSL and may potentially also change in shape.'*

3. Comment by Referee:
*At the end of the first paragraph in Section 3.1 you mention that the long-term trend from the observations was removed. Was it a linear trend and was the residual of the fit tested for being mostly white noise?*

Response:
Yes, a linear trend. The residual is not entirely white noise though, as it includes cyclicity, e.g. the annual cycle. The corresponding ESL in terms of annual maxima however have been tested positively for being white noise. An explanation has been added in the text (p. 8, line 2).'

4. Comment by Referee:
*In the caption of Figure A2 you mention that "The respective long-term mean has been removed for both time series." and you show the MHW as a dashed line in the figure. The information from the figure and the caption might become clearer if either both the long-term mean and the MHW would be indicated in the figure or was it anyway the MHW which was removed as implied in the text in Section 3.1?*

Response:
In this Figure (now Fig. A1) only the long-term mean has been removed, it is thus the y=0 line. It has now been added to the Figure. In Section 3.1, we compare values respective to the MHW (i.e. the dashed line in Figure A.1.). Additionally, we also changed the solid line to dots in the observation-based figure.

5. Comment by Referee:
*I am not against a conservative approach to show only the 50-year return values from a 50 year long record. Others calculate up to three-fold return periods given the length of a time series. Are there reasons to not calculate the 100-year return level at Husum & Norderney?*

Response:
Yes, we want to only show the non-parametric estimates based on observations, otherwise assumption have to be made, e.g. about the choice of the fitted extreme value distribution. At this stage of the model-data comparison we prefer to stick to non-parametric statistics.

6. Comment by Referee:
*In the last paragraph of Section 3.1 you argue that the model is spatially coherent along the German Bight coast and in Figure A7 you show the average along the coast. I think you need to show the median with interquartile range or something similar to support that claim. It would allow the reader to assure herself that local effects and such are not playing an important role.*

Response:
The figure has been adjusted to show the range among German Bight points rather than the mean only. Note that due to a rearrangement of the Supplementary Material, the Figure is now Fig. A6.

7. Comment by Referee:
*On Page 12 you argue that the pattern of the composites are robust against minor changes in the threshold of what constitutes "high ESL". It might help the reader to integrate "minor changes" into the story a little better if those changes could be related to a number. For example mean plus 1.5 standard deviations plus/minus 0.25 standard deviations, or 80 to 120 periods of "high ESL".*

Response:

Due to the high fluctuations in ESL, the corresponding SLP pattern also shows quite a large variance, which manifests in changes in spatial extent and strength of individual years of the composite. However, the wider spatial characteristics of the associated SLP anomaly averaged over periods of enhanced ESL remains similar if the threshold value changes. We adjusted the text accordingly, toning down some statements, and now explain the term 'minor changes' in more detail (p. 13, line 27f.):

*"… minor changes in its value, specifically the range of plus/minus 0.25 standard deviations around the chosen threshold. Yet, in the case of the ESL composites, the spatial variability of associated SLP patterns is large and single years can differ in shape and magnitude. The broader spatial character of the mean anomaly pattern, however, remains robust."*

8. Comment by Referee:
   *In the discussion on page 17 the non-parametric estimates of return levels are compared to various estimates from fitting distributions with certain parameters to the sampled distributions. I think this discussion would become clearer if the non-parametric estimates had been introduced in a sentence or two. How did you exactly determine the non-parametric estimates? This could also be added to the method part.*

   Response:
   The non-parametric estimates were inferred via an empirical cumulative distribution. A paragraph has been added in the methods section (p. 7, lines 10f.):
   *"These non-parametric estimates have been inferred by first ranking the data points of the sea level time series and associating a cumulative probability to each value. The probability of exceedance is P = m/(N+1), where m is the rank of N observations ordered in decreasing order. Following Eq. 2, return periods are again defined as the reciprocal of the respective probability of exceedance."*

9. Comment by Referee:
   *On page 19, line 29 you mention that the GCM does not account for melting ice sheets. Is the global mean thermosteric effect included in the analyzed sea level?*

   Response:
   No, as a Boussinesq model, the thermosteric effect is not accounted for and not included in the analysis. However, even though the thermosteric effect is also prognostically calculated by the model and could in theory be added linearly, this is problematic as the parent GCM exhibits considerable drift in the global thermosteric sea level; with the effect from the deep ocean, the 100-year spin-up used in our setup would be too short to account for this. We therefore rather focus on changes in dynamics.
   We now briefly address this issue in the discussion (p. 21, line 4f.):
   *"Furthermore, a transient sea level rise due to the melting of ice sheets, post-glacial isostatic rebound or the thermosteric effect is not accounted for in the model and a potential increase in ESL with a gradual rise in the BSL base could not be investigated. Such transient sea level changes can further impact ESL on longer time scales, since the sea level distribution shifts with changes in BSL and may potentially also change in shape."*

All technical corrections have been incorporated.

---

## Author Comment (AC3) · 15 May 2019

**The long-term variability of extreme sea levels in the German Bight**

**Andreas Lang and Uwe Mikolajewicz**

**Authors' response**

**RC2 Review by Anonymous Referee #2**

We thank Anonymous Referee #2 for the helpful comments. Responses to the individual comments are listed below. Page and line numbers in the responses refer to the updated version of the manuscript; changes therein are marked in red.

1. Comment by Referee:
   *My biggest objection to the manuscript is the structuring, where I think the supplementary material is much to extensive. Parts of it like the comparison to observations would be better put in the text, and other parts, I think, could be dispensed of. Specifically I think figs. A2,A3,A7 would be fit better as part of the article. Figs A1,A9 and A10 I believe did not add much and could likely be dispensed of. However, I think if would be nice to add quantile plots of model vs observations for different 100 year periods, too have a more direct comparison to observations*

   Response:
   We agree with the referee that the supplementary material is very extensive. We therefore decided to follow the referee's suggestion of leaving out two of the specified figures (A1 & A9). However, we believe that the other figures give useful additional information to the manuscript and should therefore be kept in the supplementary material. To not disturb the reader's flow with additional, yet not vital material for the analysis, we prefer not to move any more figures into the main text, since the main focus of the manuscript is the understanding of long-term variability of ESL rather than the evaluation of model results against the observational record.
   Yet, we did add the suggested quantile plot of model vs observation for each century (Fig. 4, see comment 4).

2. Comment by Referee:
   *The 16 m minimum depth. This is discussed a bit, but not in relation to the representation of bottom topography. Being myself unfamiliar with the German Bight, I don't know if representation of complex bottom topography is a problem in the area. However, I think that it should be stated in the article if this is a potential problem or if the bottom is essentially flat.*

   Response:
   We apologize, the expression '16 m minimum water depth' is imprecise and probably misleading. It rather refers to the uppermost layer thickness (see page 5, line 11), the effective water depth is ultimately dependent on tide, surge etc. and can thus result in lower water depths in the German Bight. This layer thickness is needed in order to prevent grid-points to become 'dry', which is not permitted in a climate model.
   Statements on page 9, line 7 and on page 20, line 27 have been rephrased. Further, a clarifying statement has been included in the Methods (page 5 line 9f):

*'In order to prevent ocean grid-points to fall dry due to strong tidal sea level variations, as for example in the English channel, MPIOM's uppermost layer thickness is set to 16 meter.'* This means, however, that rather minimum than maximum sea levels are subject to this feature, as the tidal coastline changes in the Wadden Sea are not represented in the model. At the main study location Cuxhaven, the sea is deeper, which increases our confidence in the results at this location compared to other points along the Wadden Sea. Besides the shallowness of the shelf, the bottom topography in the German Bight is rather smooth (see also Fig. 1) though. During surge events, the sea level rises rather uniform in the region, and the influence of bottom topography plays a subordinate role compared to the rather complex horizontal coastline geometry. A sentence about this has been added in the discussion (page 20, line 29f):

*'At Cuxhaven, this effect [lower wind surges due to simplifications in the model bathymetry] should be smaller than at points along the flatter Wadden Sea where the tidal oscillations and the shallow waters lead to coastline changes which cannot be represented here. The influence of the bottom topography is expected to play a subordinate role compared to the rather complex horizontal coastline geometry.'*

3. Comment by Referee:
*Sect. 3 how mean high water is defined should be in the paper.*

Response:
A definition has been included (time-mean over tidal maximum values; p. 8, line 6)

4. Comment by Referee:
*Sect. 3.1 row 25 comparison to observations. It is stated that ESL relative to MHW compare well to observations, but to me it seems that ESL is likely somewhat overestimated. I think the suggested quantile plots would be helpful here.*

Response:
The main point is that one realization (i.e. the observation-based record) lies within the range of simulated ESL, thus "compares well". The added quantile-quantile plot (new Fig. 4, see Comment 1) shows that for most sea levels, observed and simulated ESL agree well, and that the bulk of the disagreement between both is related to the very upper end of the ESL distribution, which is subject to substantial variability.

[Figure]

*Figure 4: Quantile-quantile plot of 100-year segments of simulated ESL at Cuxhaven against the 100-year long ESL from the tide gauge record. Colors following the gradient from light red to black represent ascending 100-year segments from 1000-2000.*

5. Comment by Referee:
   *Fig 4. I was first confused over how the return levels were estimated here. I think it should be stated in the caption that they are non-parametric, and I also suggest that you add a second panel showing there parametric counterparts for the same time period. Also a non-parametric return level curve for the whole data set would be interesting.*

   Response:
   The specification 'non-parametric' has been added in the Figure caption (now Figure 5). Additionally, a short description about how they were inferred has been added in the method section (p. 7, lines 9f.). However, we decided not to show the full parametric counterparts due to length reasons; the corresponding parametric 1000-year return level estimates are already shown in the bars on the right. The non-parametric 1000-year return level is also indicated in the figure, a curve for the whole data set though is also left out for length reasons.

6. Comment by Referee:
   *Page 10 just before Sect. 3.2, it is stated that other grid points are similar to Cuxhaven. I don't dispute this, but the figure shows a mean of the other gridpoints, and says noting about the spread, so I would rephrase.*

   Response:
   We modified the figure which now includes the range of other gridpoints along the German Bight, and thus shows the spread rather than the mean only.

7. Comment by Referee:
   *What about the correlation of the running mean series from Fig 4. It looks like it might be higher?*

   Response:
   This only appears to be higher. We double-checked and got the same value.

8. Comment by Referee:
   *Fig. 9 and all wavelet figures. Please state which series leads which for a given arrow direction, not just the first series.*

   Response:
   The respective captions have been adjusted.

9. Comment by Referee:
   *Sect. 3.4. I think you could develop the last thoughts in this paragraph more, and also show the SLP pattern when BSL has been removed as an additional panel in Fig. 8*

   Response:
   We added a couple of lines to this paragraph (p. 17, lines 13f.). For length and readability reasons, however, we prefer not to explicitly show the Figure in the mauscript. Due to the high fluctuations in ESL, the corresponding SLP pattern also shows quite a large variance, which manifests in changes in spatial extent and strength of individual years of the composite. Due to this large variability, the SLP pattern where BSL has been removed is not identical to the one associated with high ESL. However, the spatial character with a tendency for a rather clockwise spinned dipole remain similar (see Fig. below).

[Figure]

*SLP composite over times of enhanced ESL after the reduction of BSL*

Lines added (p. 17, lines 13f.):

*"[...] the large-scale circulation pattern associated with high ESL qualitatively persists if annual median is subtracted, although weaker. The comparably large spatial variance in the ESL pattern, however, leads to slight shifts in the location of the centers of action of the corresponding SLP dipole; yet, both ESL and ESL-BSL share a tendency towards a less meridional character of the associated SLP patterns whose centers turn clockwise compared to the BSL-related SLP anomaly. This similarity emphasizes that the SLP pattern associated with high ESL is linked to the surge residual variations. BSL variations are of much smaller amplitude than ESL variations and thus become marginal amid the strong variability of the latter."*

10. Comment by Referee:

*Page 19 around line 30. You mention the lack of melting ice-sheets, but not the thermosteric effect. I assume your model is Boussinesq so you don't have this either?*

Response:

Yes, it is Boussinesq. Even though the thermosteric effect is calculated by the model prognostics and could in theory be added linearly to the sea level time series, this is problematic as the global model exhibits considerable drift; with the effect from the deep ocean, the 100-year spin-up used in our setup is too short to account for this. We therefore rather focus on changes in dynamics.

A clarifying sentence has been added in the methods (p. 6, line 1-2), and we now included the omission of the thermosteric effect also in the discussion (p. 21, line 5).

All technical corrections have been incorporated.